# The MAP kinase pathway coordinates crossover designation with disassembly of synaptonemal complex proteins during meiosis

Saravanapriah Nadarajan[1], Firaz Mohideen[2], Yonatan B Tzur[1†], Nuria Ferrandiz[3], Oliver Crawley[3], Alex Montoya[4], Peter Faull[4], Ambrosius P Snijders[4‡], Pedro R Cutillas[4§], Ashwini Jambhekar[1,5], Michael D Blower[1,5], Enrique Martinez-Perez[3], J Wade Harper[2], Monica P Colaiacovo[1*]

[1]Department of Genetics, Harvard Medical School, Boston, United States; [2]Department of Cell Biology, Harvard Medical School, Boston, United States; [3]MRC Clinical Sciences Centre, Imperial College London, London, United Kingdom; [4]Proteomics facility, MRC Clinical Sciences Centre, Imperial College London, London, United Kingdom; [5]Department of Molecular Biology, Massachusetts General Hospital, Boston, United States

*For correspondence:
mcolaiacovo@genetics.med.
harvard.edu

Present address: [†]Department of Genetics, Alexander Silberman Institute of Life Sciences, Hebrew University of Jerusalem, Jerusalem, Israel; [‡]London Research Institute, South Mimms, United Kingdom; [§]Barts Cancer Institute, London, United Kingdom

Competing interests: The authors declare that no competing interests exist.

**Abstract** Asymmetric disassembly of the synaptonemal complex (SC) is crucial for proper meiotic chromosome segregation. However, the signaling mechanisms that directly regulate this process are poorly understood. Here we show that the mammalian Rho GEF homolog, ECT-2, functions through the conserved RAS/ERK MAP kinase signaling pathway in the *C. elegans* germline to regulate the disassembly of SC proteins. We find that SYP-2, a SC central region component, is a potential target for MPK-1-mediated phosphorylation and that constitutively phosphorylated SYP-2 impairs the disassembly of SC proteins from chromosomal domains referred to as the long arms of the bivalents. Inactivation of MAP kinase at late pachytene is critical for timely disassembly of the SC proteins from the long arms, and is dependent on the crossover (CO) promoting factors ZHP-3/RNF212/Zip3 and COSA-1/CNTD1. We propose that the conserved MAP kinase pathway coordinates CO designation with the disassembly of SC proteins to ensure accurate chromosome segregation.

## Introduction

Accurate chromosome segregation during meiosis is critical for sexually reproducing organisms. Meiosis is the specialized cell division program by which diploid germ cells generate haploid gametes that during fertilization will form a diploid zygote. This is accomplished by following a single round of DNA replication with two consecutive rounds of cell division (meiosis I and II) in which, first pairs of homologous chromosomes (bivalents), and then sister chromatids, separate away from each other. At center stage during prophase I of meiosis from yeast to humans is a tripartite proteinaceous structure known as the synaptonemal complex (SC) (*Colaiácovo, 2006*). The SC is a ladder-like structure comprised of lateral element proteins running along the axes of the homologs and central region proteins connecting these axes (*Colaiácovo, 2006*; *Page and Hawley, 2003*). The SC is required to stabilize the interactions between pairs of homologous chromosomes and for interhomolog crossover (CO) formation in budding yeast, worms, flies, and mice (*Nag et al., 1995*; *Storlazzi et al., 1996*; *Page and Hawley, 2001*; *MacQueen et al., 2002*; *Colaiácovo et al., 2003*;

**eLife digest** Most plants and animals, including humans, have cells that contain two copies of every chromosome, with one set inherited from each parent. However, reproductive cells (such as eggs and sperm) contain just one copy of every chromosome so that when they fuse together at fertilization, the resulting cell will have the usual two copies of each chromosome.

Embryos that have incorrect numbers of chromosome copies either fail to survive or develop disorders such as Down syndrome. Therefore, it is important that when cells divide to form new reproductive cells, their chromosomes are correctly segregated.

To end up with one copy of each chromosome, reproductive cells undergo a form of cell division called meiosis. During meiosis, pairs of chromosomes are held together by a zipper-like structure called the synaptonemal complex. While held together like this, each chromosome in the pair exchanges DNA with the other by forming junctions called crossovers. Once DNA exchange is completed, the synaptonemal complex disappears from certain regions of the chromosome.

Using a range of genetic, biochemical and cell biological approaches, Nadarajan et al. have now investigated how crossover formation and the disassembly of the synaptonemal complex are coordinated in the reproductive cells of a roundworm called *Caenorhabditis elegans*. This revealed that a signaling pathway called the MAP kinase pathway regulates the removal of synaptonemal complex proteins from particular sites between the paired chromosomes. Turning off this pathway's activity is required for the timely disassembly of this complex, and depends on proteins that are involved in crossover formation.

This regulatory mechanism likely ensures that the synaptonemal complex starts to disassemble only after the physical attachments between the paired chromosomes are "locked in", thus ensuring that reproductive cells receive the correct number of chromosomes. Given that the MAP kinase pathway regulates cell processes in many different organisms, a future challenge is to determine whether this pathway regulates the synaptonemal complex in other species as well.

*de Vries et al., 2005*; *Smolikov et al., 2007a*; *2007b*; *2009*), all of which are prerequisite steps for achieving accurate chromosome segregation at meiosis I. Importantly, the SC must be disassembled prior to the end of prophase I to ensure timely and proper chromosome segregation (*Sourirajan and Lichten, 2008*; *Jordan et al., 2009*; *Dix et al., 1997*). Several proteins have been recently implicated in regulating the disassembly of SC proteins, including polo-like kinase in mammals and budding yeast, Ipl1/Aurora B kinase in budding yeast, the condensin complex component dcap-g in flies and both AKIR-1 (a member of the akirin protein family) and ZHP-3 (ortholog of budding yeast Zip3) in worms (*Sourirajan and Lichten, 2008*; *Jordan et al., 2009*; *Resnick et al., 2009*; *Jordan et al., 2012*; *Clemons et al., 2013*; *Bhalla et al., 2008*). However, whether these factors directly regulate the SC in vivo and the mechanisms by which they promote the disassembly of SC proteins are not fully understood.

We previously showed that the SC disassembles asymmetrically in *C. elegans*, progressing from being localized along the full length of the interface between homologous chromosomes in early prophase to persisting at discrete chromosome locations, termed the short arm domains, corresponding to one end of every pair of homologs at late prophase I (*Figure 1A*) (*Nabeshima et al., 2005*). Importantly, factors required for CO formation are necessary for the asymmetric disassembly of the SC and localized retention of SC central region proteins. Since in *C. elegans,* a single CO occurs at the terminal third of every pair of homologous chromosomes, we proposed that chromosomes remodel around the single off-centered CO event at the pachytene-diplotene transition. This results in bivalents with a cruciform configuration comprised of two perpendicular chromosomal axes (namely the long and short arm domains) intersecting at the chiasma, where the long arms face the poles and the short arms occupy an equatorial position on the metaphase plate (*Figure 1A*) (*Nabeshima et al., 2005*; *Riddle et al., 1997*; *Maddox et al., 2004*). This remodeling includes changes in chromosome compaction as well as changes in both the localization and the types of proteins associated with the long and short arm domains (*Figure 1A*) (*Nabeshima et al., 2005*; *Chan et al., 2004*; *de Carvalho et al., 2008*; *Martinez-Perez and Villeneuve, 2005*). Subsequent

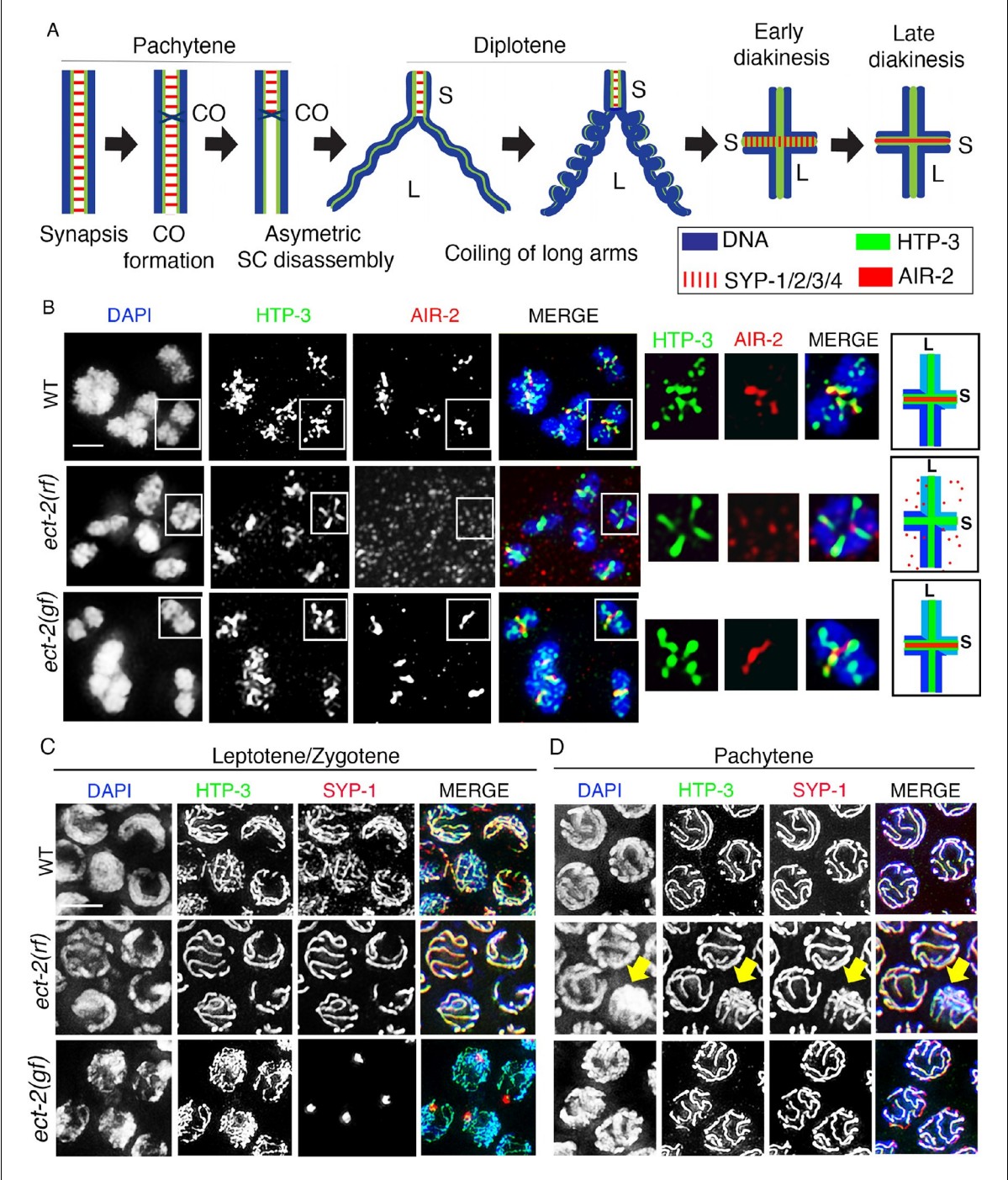

**Figure 1.** ECT-2 regulates AIR-2 localization and SC dynamics in meiotic prophase I. (**A**) Schematic representation of SC dynamics and chromosome remodeling during prophase I of meiosis. A single pair of homologous chromosomes (bivalent) is shown for simplicity. Upon entrance into pachytene, the SC is present along the full length of the pairs of homologous chromosomes. CO formation is completed within the context of fully synapsed chromosomes, and in worms, a single CO is formed per homolog pair usually at an off-centered position. Chromosome remodeling has been proposed to take place around the off-centered CO (or CO precursor) resulting in a cruciform configuration comprised of two intersecting perpendicular chromosomal axes of different lengths (long and short arms of the bivalent, corresponding to the longest and shortest distances from the off-centered CO/CO precursor site to opposite ends of the chromosomes). This remodeling involves disassembly of central region components of the SC (SYP-1/2/3/4) along the long arms of the bivalents starting during late pachytene and diplotene resulting in the restricted localization of these proteins to the short arms. During diplotene and diakinesis, chromosomes undergo condensation as evidenced by a coiling of the arms and increased bivalent compaction. In late diakinesis, the SC proteins located on the short arms are replaced by AIR-2, which promotes the separation of the homologs at the end of meiosis I. CO – crossover, S – short arm and L – long arm. (**B**) Immunolocalization of HTP-3 and AIR-2 on -1 oocytes at

*Figure 1 continued on next page*

*Figure 1 continued*

diakinesis in wild type, *ect-2(gf)* and *ect-2(rf)* gonads. AIR-2 is localized to the short arm of the bivalents in wild type and *ect-2(gf)* mutants, but fails to localize onto chromosomes in *ect-2(rf)* mutants. Diagrams on the right illustrate the cruciform structure of the bivalents at this stage consisting of long (L) and short (S) arms and the localization of AIR-2 (red) and HTP-3 (green) in wild type and *ect-2(rf)* mutant. White box indicates the bivalent shown at a higher magnification on the right. Bivalents with both long and short arms clearly displayed were chosen for higher magnification. (C) Immunolocalization of HTP-3 and SYP-1 on leptotene/zygotene nuclei from gonads of the indicated genotypes. SYP-1 aggregates (polycomplexes) are detected in *ect-2(gf)* mutants. (D) SYP-1 and HTP-3 localize throughout the full length of the synapsed homologous chromosomes during pachytene in wild type and in most pachytene nuclei in *ect-2(rf)* mutants. Arrowhead indicates a nucleus where chromosomes persist in the DAPI-bright and tighter clustered configuration characteristic of the leptotene/zygotene stage in *ect-2(rf)*. In *ect-2(gf)* mutants, meiotic nuclei that progressed through the leptotene/zygotene stage before the shift to the non-permissive temperature show wild type-like SYP-1 localization. Worms from all the indicated genotypes, including wild type, were grown at 15°C, shifted to 25°C at the L4 stage, and analyzed 18–24 hr post-L4. n>26 gonads were examined for each genotype in (B) and n>15 in (C) and (D). Bars, 2 μm.

The following figure supplements are available for figure 1:

**Figure supplement 1.** *ect-2(RNAi)* affects AIR-2::GFP loading on the chromosomes.

**Figure supplement 2.** *ect-2(e1778)* null mutant exhibits disorganized germline with fewer germ cells.

**Figure supplement 3.** Rescue of *ect-2(ax751rf)* phenotypes by expression of functional ECT-2::GFP in the germline.

**Figure supplement 4.** Co-localization of ECT-2::GFP and RHO-1 in the germline.

---

studies in yeast, flies and mice (*Newnham et al., 2010*; *Qiao et al., 2012*; *Bisig et al., 2012*; *Takeo et al., 2011*; *Gladstone et al., 2009*) also observed an asymmetric disassembly of the SC, with a residual localized retention of the SC at centromeres. However, the mechanism linking CO recombination sites to asymmetric SC disassembly remained unknown.

Recently, a two-step CO specification process has been described to take place following SC assembly as prospective CO sites progressively differentiate during *C. elegans* and mouse meiosis (*Yokoo et al., 2012*; *Holloway et al., 2014*). This consists of CO licensing during mid-pachytene followed by CO designation at or just prior to the mid-to-late pachytene transition in a manner dependent of the pro-CO factor COSA-1/CNTD1. Since DSBs outnumber COs in most species (*Martinez-Perez and Colaiácovo, 2009*) this has been proposed as a strategy to pare down the number of early recombination sites that will become CO sites thus both ensuring and limiting the number of COs. These and other features of meiosis in *C. elegans*, including the existence of various markers that distinguish the short and long arm subdomains and the CO precursor sites, therefore provide an ideal scenario to understand the regulation of the disassembly of the SC proteins from distinct chromosome subdomains and late prophase I chromosome remodeling.

Here, we report that regulation of the disassembly of the SC proteins from the long arms of the bivalents in *C. elegans* requires the mammalian Rho GEF homolog, ECT-2. We show that ECT-2 functions through the conserved MAP kinase pathway to regulate the asymmetric disassembly of SC proteins during prophase I of meiosis. We show that MPK-1 potentially directly phosphorylates SYP-2, a central region component of the SC, and that constitutively phosphorylated SYP-2 impairs the disassembly of SC proteins from the long arms. Moreover, inactivation of MPK-1 takes place in late pachytene in a manner dependent on pro-CO factors ZHP-3/RNF212/Zip3 and COSA-1/CNTD1 and concomitant with the initiation of SC disassembly. Therefore, we propose a model in which MPK-1 is inactivated in response to CO designation resulting in either de novo loading of unphosphorylated SYP-2 or dephosphorylation of chromatin-associated SYP-2, which triggers disassembly of SC proteins from along the long arms. Thus, coordination between CO designation and the disassembly of SC proteins executed via a conserved MAP kinase pathway is critical for ensuring accurate chromosome segregation during meiosis.

## Results

### ECT-2 regulates synaptonemal complex dynamics

We identified *ect-2*, the homolog of mammalian Rho GEF, in a targeted RNAi screen for novel components regulating chromosome remodeling and short/long arm identity by using the mislocalization of Aurora B kinase, AIR-2, which localizes to the short arms of diakinesis bivalents in wild type oocytes as a readout (see Materials and methods). ECT-2 (Epithelial Cell Transforming sequence 2) is a highly conserved protein, initially identified as a proto-oncogene in cell culture (*Miki et al., 1993*). It encodes a Guanine nucleotide Exchange Factor (GEF) that belongs to the Dbl family and functions as a key activator of Rho GTPase mediated signaling with roles during cytokinesis, DNA damage-induced cell death, cell polarity establishment during embryogenesis, vulval development, and epidermal P cell migration (*Prokopenko et al., 1999*; *Saito et al., 2003*; *Morita et al., 2005*; *Srougi and Burridge, 2011*; *Canevascini et al., 2005*; *Motegi and Sugimoto, 2006*). Although ECT2 is expressed in testis and ovaries in humans (*Hirata et al., 2009*; *Fields and Justilien, 2010*), understanding its role during mammalian meiosis is challenging due to its earlier roles in development. The availability of conditional mutants, such as temperature-sensitive mutants in *C. elegans,* therefore provided a unique opportunity to discover a novel role for ECT-2 in meiosis.

Analysis of *ect-2(ax751rf)* temperature-sensitive reduction-of-function mutants and of worms depleted of *ect-2* by RNAi revealed a failure of AIR-2 to load on the chromosomes in late diakinesis even though AIR-2 is present inside the nucleus (*Figure 1B* and *Figure 1—figure supplement 1*). Importantly, we shifted *ect-2(ax751rf)* worms to the non-permissive temperature at the L4 larval stage to bypass the requirements for ECT-2 during somatic development and germ cell mitotic proliferation such as seen in *ect-2(e1778)* null mutants, which are sterile and exhibit fewer germ cells with abnormal nuclei (*Figure 1—figure supplement 2*). *ect-2(ax751rf)* mutants also exhibited a reduced brood size, increased embryonic lethality and a High Incidence of Males (Him) at the non-permissive temperature, all phenotypes indicative of increased meiotic chromosome nondisjunction (*Table 1*).

To investigate the localization of ECT-2 in the germline, we utilized a ECT-2::GFP transgene driven by the *ect-2* promoter (*Chan and Nance, 2013*). ECT-2::GFP is able to rescue the reduced brood size, embryonic lethality and Him phenotypes observed in the *ect-2(ax751rf)* mutants (*Figure 1—figure supplement 3*), confirming that mutation of *ect-2* impairs fertility. We found that ECT-2::GFP localizes throughout the germline from the mitotically dividing nuclei at the premeiotic tip, where it is enriched at the germ cell membrane, to the end of diakinesis, where it exhibits a stronger nuclear signal (*Figure 1—figure supplement 4*).

Importantly, we discovered a novel role for ECT-2 in regulating SC dynamics, a term we will use herein to refer to SC assembly and disassembly. In *ect-2(ax751rf)* mutants, SC formation is indistinguishable from wild type in most of the meiotic nuclei except for a few pachytene nuclei (11.1%, n=120/1080 pachytene nuclei) which display reduced SC assembly (*Figures 1C and D*). SC disassembly occurred as in wild type, initiating in late pachytene and resulting in the restricted localization of the central region proteins of the SC, as exemplified by SYP-1, to the short arm of the bivalents (*Figure 2* and *Figure 2—figure supplement 1*). In contrast, *ect-2(zh8gf)* gain-of-function mutants, in which a mutation in an auto-inhibitory BRCT domain retains ECT-2 in its constitutively active configuration (*Canevascini et al., 2005*), exhibited defects in both SC assembly and disassembly at the non-permissive temperature. Specifically, the SC failed to assemble on the chromosomes and instead SYP-1 formed aggregates referred to as polycomplexes (*Figure 1C*). Those nuclei that had already gone through the leptotene/zygotene stage before the temperature shift showed normal synapsis during early and mid pachytene stages, but the central region components of the SC remained localized along the long arms of the bivalents and failed to become restricted to the short arms during late prophase (a defect herein referred to as impaired disassembly of the SC proteins) (*Figure 2*). Importantly, the HORMA domain containing lateral element protein HTP-3, exhibits a wild type pattern of localization in *ect-2* mutants (*Figures 1C,D* and *2*). This suggests that ECT-2 acts in a specific manner to regulate SC dynamics, which involves regulating central region components of the SC.

**Table 1.** Worms were maintained at 15°C and then shifted to 25°C at the L4 larval stage. All the analyses were conducted at 25°C for all the genotypes indicated above. The 'Eggs Laid' column indicates the average number of eggs laid (including both hatched and non-hatched embryos) per P0 hermaphrodite ± standard deviation. % Embryonic lethality was calculated by dividing the number of non-hatched embryos by the total number of hatched and non-hatched embryos laid. % Males was calculated by dividing the total number of males observed by the total number of hatched (viable) progeny scored. N = total number of P0 worms for which entire broods were scored. N.A.= not applicable.

| GENOTYPE | EGGS LAID | % EMBRYONIC LETHALITY | % MALES | N |
|---|---|---|---|---|
| Wild type | 192 ± 32.3 | 0 | 0 | 20 |
| ect-2(ax751) | 109 ± 38.6 | 76.4 | 3.25 | 30 |
| ect-2(zh8) | 4.5 ± 6.7 | 93.4 | 0 | 32 |
| let-60(ga89) | 3.1 ± 5.6 | 83.8 | 0 | 20 |
| mpk-1(ga111) | 5.8 ± 7.6 | 56 | 0 | 20 |
| ect-2(ax751); let-60(ga89) | 0.7 ± 0.7 | 100 | 0 | 20 |
| ect-2(zh8); mpk-1(ga111) | 0 | N.A. | N.A. | 30 |

## ECT-2 specifically regulates the disassembly of SC proteins and is not required for other aspects of late prophase chromosome remodeling

Since the SC is no longer restricted to the short arms of the bivalents in *ect-2(zh8gf)* mutants, we tested whether late prophase chromosome remodeling is altered in this mutant. First, we observed that AIR-2 localization is still successfully restricted to the short arms of the bivalents at diakinesis in *ect-2(zh8gf)* mutants (*Figure 1B*). Second, LAB-1, a proposed functional ortholog of Shugoshin, is still lost from the short arms and restricted to the long arms of the bivalents as in wild type (*de Carvalho et al., 2008*; *Schvarzstein et al., 2010*; *Tzur et al., 2012*) (*Figure 3*). These data indicate that key aspects of late prophase chromosome remodeling, namely LAB-1 and AIR-2 restricted localizations, are not altered in *ect-2(zh8gf)* mutants and that ECT-2 specifically regulates the disassembly of the SC proteins during prophase I of meiosis.

## The ERK MAP kinase pathway is involved in regulating SC dynamics

ECT-2 has been shown to activate the RAS/MAP kinase signaling cascade to promote primary vulval cell fate specification during vulval development in *C. elegans* (*Canevascini et al., 2005*). The RAS/ERK (Extracellular signal-regulated kinase) MAP Kinase signaling regulates various aspects of the cell cycle and components of the signaling cascade are highly conserved between *C. elegans* and mammals (*Sundaram, 2013*). In *C. elegans*, the ERK MAP kinase, MPK-1, is expressed in both somatic tissues as well as in the germline where it controls several aspects of germ line development including pachytene progression and germ cell survival (*Church et al., 1995*; *Kritikou et al., 2006*; *Lee et al., 2007*). However, the role of MPK-1 in SC dynamics has never been explored. We therefore tested whether the ERK MAP kinase pathway regulates SC dynamics in the germline. Analysis of *mpk-1 (ga117)* null, *mpk-1(ga111lf)* and *mpk-1(ku1lf)* temperature-sensitive loss-of-function mutants revealed defects in SC assembly as indicated by the formation of polycomplexes, similar to *ect-2 (zh8gf)* mutants at the non-permissive temperature (*Figure 1C* and *Figure 2—figure supplement 3*). However, germline nuclei that had already passed through the leptotene/zygotene stage before the temperature shift exhibited normal SC tracts along chromosomes at the pachytene stage (*Figure 2— figure supplement 3*). Since nuclei fail to progress from early/mid-pachytene to late-pachytene in *mpk-1(ga117)* null mutants (*Lee et al., 2007*), we analyzed the disassembly of SC proteins in *mpk-1 (ga111lf)* mutants. The *mpk-1(ga111)* temperature sensitive loss-of-function mutant has been shown to still have a low level of dpMPK-1 activity and an incompletely penetrant pachytene arrest phenotype (*Lee et al., 2007*). Nuclei that were able to progress from pachytene to diplotene exhibited normal SC disassembly similar to the *ect-2(ax751rf)* mutant (*Figure 2*).

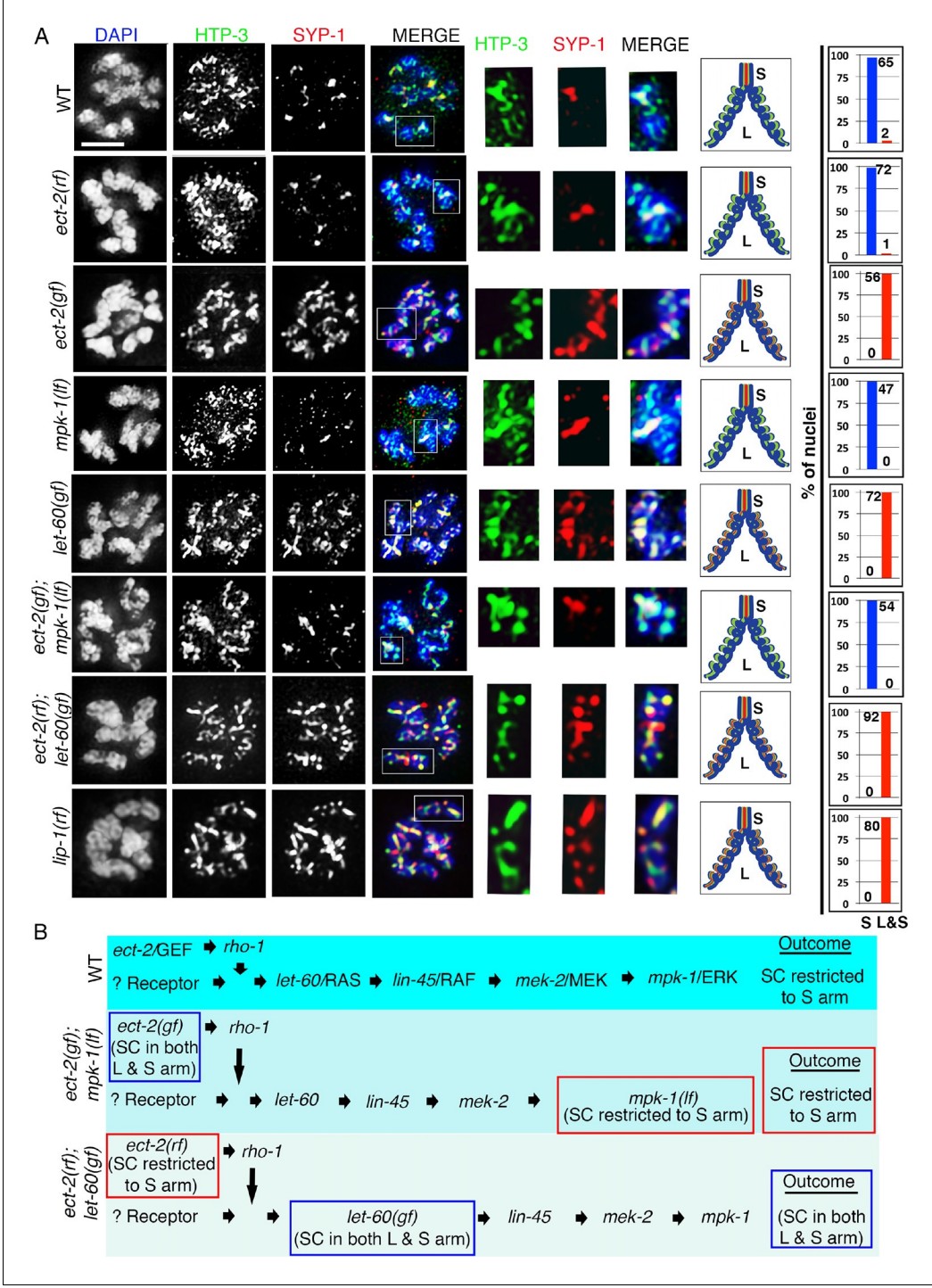

**Figure 2.** ECT-2 regulates the disassembly of SC proteins from the long arms of the bivalents through the MPK-1 pathway. Co-staining with HTP-3 (green), SYP-1 (red) and DAPI (blue) of diplotene and early diakinesis nuclei from the indicated genotypes. At diplotene and early diakinesis, SYP-1 localization is restricted to the short arm in both wild type and *ect-2(rf)* mutants. In contrast, SYP-1 fails to disassemble from the long arm and become restricted to the short arm of the bivalents in *ect-2(gf)*, *let-60(gf)* and *lip-1(rf)* mutants. *mpk-1(lf)* mutants suppress the defect in disassembly of SC proteins from the long arms observed in *ect-2(gf)* mutants whereas *ect-2(rf); let-60(gf)* double mutants exhibit the phenotype of *let-60(gf)* mutants. Illustrations depict the bivalent configuration at this stage. White box indicates the bivalent shown at a higher magnification on the right. Bivalents with both long and short arms clearly displayed were chosen for higher magnification. Worms from all the indicated genotypes, including wild type, were grown at 15°C, shifted to 25°C at the L4 stage, and analyzed 18–24 hr post-L4. Histograms on the

*Figure 2 continued*

right indicate the percentage of diplotene and diakinesis stage nuclei with SYP-1 either only on the short arm (S, blue) or on both long and short arms (red, L&S) of the bivalents. All the bivalents were examined in every nucleus and the bivalents in the same nucleus either all exhibited SYP-1 staining on both the long and short arms or all exhibited staining only on the short arms. Numbers of nuclei scored are shown. (**B**) Schematic representation shows the crosstalk between the conserved ECT-2 and RAS/MAPK pathways and restriction of the SC to the short (S) arm of the bivalent at diakinesis in wild type. Remaining schematic shows epistasis analysis in *ect-2(gf); mpk-1 (ga111lf)* and *ect-2(rf); let-60(gf)* double mutants. S indicates short arm and L indicates long arm. n>15 gonads arms were analyzed for each genotype. Bar, 2 μm.

The following figure supplements are available for figure 2:

**Figure supplement 1.** SYP-1 and HTP-3 localization at diakinesis in *ect-2* mutants.

**Figure supplement 2.** SYP-1 localization in *unc-4* and *unc-32* mutants.

**Figure supplement 3.** SYP-1 localization in MAP kinase mutants.

**Figure supplement 4.** LAB-1 localization in *let-60(ga89gf)* and *lip-1(zh15rf)* mutants.

**Figure supplement 5.** Immunolocalization of dpMPK-1 in the germlines of *ect-2* mutants.

**Figure supplement 6.** Active dpMPK-1 level is reduced upon *rho-1* RNAi in the germline.

To further examine the role of the ERK MAP kinase pathway in the disassembly of SC proteins, we analyzed *let-60*, which encodes for the RAS protein that functions upstream of the MAP kinase cascade to activate the MAP kinase pathway. A *let-60(ga89gf)* temperature-sensitive gain-of-function mutant leads to constitutive activation of MAP kinase both in somatic tissues and the germline (***Lee et al., 2007***). Similar to the *ect-2(zh8gf)* mutant, *let-60(ga89gf)* mutants exhibit defects in both SC assembly, as evidenced by the presence of polycomplexes, and the disassembly of the SC proteins from the long arms of the bivalents, where they persisted at the non-permissive temperature (***Figure 2*** and ***Figure 2—figure supplement 3***). Similar defects in the disassembly of SC proteins are also observed in the *lip-1(zh15rf)* reduction-of-function mutant (***Figure 2***), where the LIP-1 phosphatase fails to inactivate, MPK-1 in late pachytene, which therefore persists throughout pachytene, diplotene and diakinesis (***Hajnal and Berset, 2002***; ***Rutkowski et al., 2011***), suggesting that the constitutive presence of active MPK-1 impairs the disassembly of SC proteins from the long arms of the bivalents. In contrast, we did not observe any defects in SC assembly in *lip-1(zh15rf)* (***Figure 2—figure supplement 3***). Further, we found that LAB-1 localization was not altered in either *let-60 (ga89gf)* or *lip-1(zh15rf)* mutants suggesting that MAP kinase specifically regulates the disassembly of SC proteins from the long arms of the bivalents and not other aspects of late prophase chromosome remodeling (***Figure 2—figure supplement 4***). Altogether these data suggest that the MAP kinase pathway plays an essential role in regulating the disassembly of SC proteins whereas the defects we see in SC assembly might be a secondary consequence of the role of MAP kinase in mitosis or due to several other germline functions of MPK-1.

## ECT-2 functions through the ERK MAP kinase pathway to regulate the disassembly of SC proteins from the long arms of the bivalents

To examine the connection between ECT-2 and the ERK MAP kinase pathway, we took advantage of the tightly regulated windows where the activated diphosphorylated form of MPK-1 (dpMPK-1) can be detected in the germline. In the *ect-2(ax751rf)* mutant, we see reduced dpMPK-1 signal in the mid-late pachytene region compared to wild type (***Figure 2—figure supplement 5A and B***). We next tested whether the Rho GTPase, RHO-1, activates ERK MAP kinase signaling in the germline, similar to its role in the regulation of vulval development (***Canevascini et al., 2005***). Given the highly abnormal gonads observed following strong loss of *rho-1* function, partial RNAi knockdown of *rho-1* was performed to obtain germlines with essentially wildtype morphology/organization (see Materials and Methods). Similarly to the *ect-2(ax751rf)* mutant, partial depletion of *rho-1* also results in

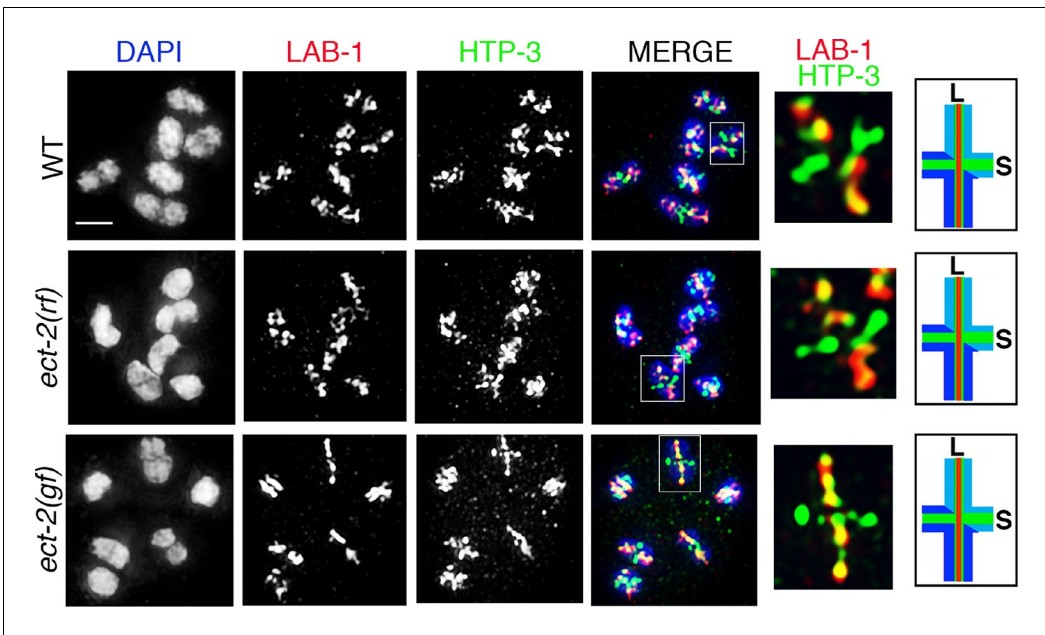

**Figure 3.** ECT-2 does not alter LAB-1 localization. Immunolocalization of LAB-1 and HTP-3 in -1 oocytes at diakinesis indicates that LAB-1 is restricted to the long arm of the bivalents in *ect-2(rf)* and *ect-2(gf)* mutants similar to wild type. Illustration depicts the cruciform structure of the bivalents at this stage and the localization of LAB-1 (red) to the long arm (L) and HTP-3 (green) to both long and short (S) arms in wild type. White box indicates the bivalent shown at a higher magnification on the right. Bivalents with both long and short arms clearly displayed were chosen for higher magnification. Worms from all the indicated genotypes, including wild type, were grown at 15°C, shifted to 25°C at the L4 stage, and analyzed 18–24 hr post-L4. n>15 gonads were analyzed for each genotype. Bar, 2 µm.

reduced dpMPK-1 signal in the mid-late pachytene region (*Figure 2—figure supplement 6*). In addition, the expression pattern of RHO-1 is similar to that observed for ECT-2 in the germline (*Figure 1—figure supplement 4*). In contrast to *ect-2(ax751rf)* mutants and to *rho-1* partial knock down, dpMPK-1 expression is not turned off at late pachytene and during diplotene in the *ect-2(zh8gf)* mutant. Instead, dpMPK-1 persists from pachytene to diakinesis similar to *let-60/RAS* gain-of-function mutants (*Figure 2—figure supplements 5A and B*; [*Lee et al., 2007*]). Taken together, these data indicate that ECT-2 and RHO-1 either promote MPK-1 activation or block MPK-1 inactivation in the germline.

Next, we performed epistasis analysis to test whether ECT-2 functions through the MAP kinase pathway to regulate the disassembly of SC proteins. First, we analyzed the brood size in *ect-2 (ax751rf); let-60(ga89gf)* double mutants (*Table 1*). *let-60(ga89gf)* mutants exhibit a severely reduced brood size compared to *ect-2(ax751rf)*. Interestingly, *ect-2(ax751rf); let-60(ga89gf)* double mutants exhibit a severely reduced brood size similar to *let-60(ga89gf)*. Further, *ect-2(ax751rf); let-60(ga89gf)* double mutants exhibit defects in the disassembly of SC proteins similar to *let-60(ga89gf)* mutants (*Figure 2A and B*). These data show that LET-60 functions downstream of ECT-2 in the germline. We also analyzed *ect-2(zh8gf); mpk-1(ga11lf1)* double mutants for defects in the disassembly of SC proteins. We found that *mpk-1(ga111lf)* is able to suppress the SC disassembly defect of *ect-2 (zh8gf)* mutants (*Figure 2A and B*). Thus, our data demonstrates that ECT-2 functions through the MAP Kinase pathway to regulate the disassembly of the SC proteins on the long arms of the bivalents uncovering a novel mode of regulation for this structure.

## Phosphorylation of SYP-2 is dependent on the ERK MAP kinase pathway

To determine how the disassembly of the SC proteins is regulated by MPK-1, we tested whether SC components might be a direct phosphorylation target of MPK-1. First we examined the central

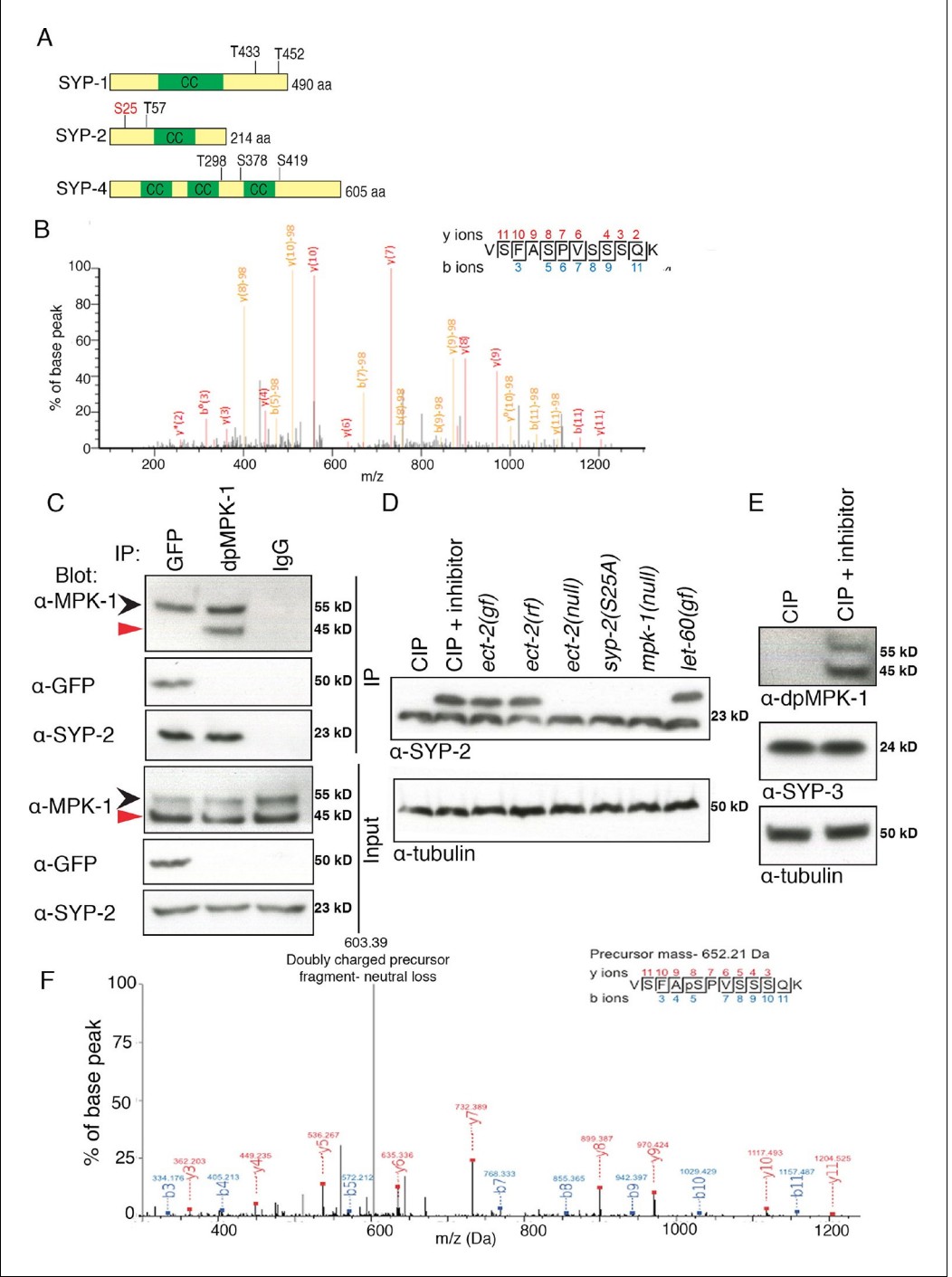

**Figure 4.** Phosphorylation of SYP-2 is dependent on the MAP kinase pathway. (**A**) Schematic representation of SYP-1, SYP-2, and SYP-4 proteins with predicted MAP kinase phosphorylation sites indicated in black and site confirmed by mass spectrometry indicated in red. CC indicates the coiled-coil domains. (**B**) MS/MS fragmentation spectrum for SYP-2 phosphopeptide VSFASPVSSSQK in the range 100–1300m/z. The annotated spectrum shows fragment ion species matched between theoretical and measured values. 'b ions' are generated through fragmentation of the peptide bond from the N-terminus, whereas 'y ions' are generated through fragmentation from the C-terminus. Ion species detected with a mass loss of 98 (phosphoric acid) are indicated in yellow; those ions without phospho-loss are annotated in red. Analysis of 'y' and 'b' ions with and without phospho-loss is consistent with phosphorylation of the second serine (VSFApSPVSSSQK), corresponding to S25 of the SYP-2 protein. (**C**) Western blots showing immunoprecipitation of SYP-2 from SYP-2::GFP whole worm lysates with a GFP antibody and dpMPK-1 and IgG immunoprecipitation from wild type whole worm lysates with dpMPK-1 and IgG

*Figure 4 continued on next page*

*Figure 4 continued*

antibodies, respectively. Standard SDS-PAGE was used and western blots were probed with the indicated antibodies. Black arrow (upper band) indicates germline-specific and red arrow (lower band) indicates soma-specific isoforms of MPK-1. The germline-specific isoform of MPK-1 co-immunoprecipitates with SYP-2 confirming the in vivo interaction detected between dpMPK-1 and SYP-2. (**D**) Calf intestinal phosphatase assay (CIP) showing SYP-2 is phosphorylated in vivo. After CIP treatment, only a faster migrating band of SYP-2 is present whereas in the presence of the phosphatase inhibitor, both faster and slower migrating bands are present. In the *mpk-1 (ga117)* and *ect-2(e1778)* null mutant lysates, the upper band is no longer present whereas in *let-60(ga89)* and *ect-2(zh8)* gain-of-function mutants, as well as in *ect-2(ax751rf)* reduction-of-function mutants, both upper and lower bands are present. In the *syp-2(S25A)* phosphodead mutant, only the lower migrating band is present. α-tubulin is used as a loading control. Worms from all indicated genotypes were grown at 15°C and shifted to 25°C at the L4 stage. Worm lysates were prepared from 18–24 post-L4 worms. Phos-tag SDS-PAGE was used for better separation and detection of phosphorylated proteins. (**E**) Western blot showing that in the presence of CIP, dpMPK-1 is not present, whereas in the absence of CIP activity dpMPK-1 is present, indicating that the CIP assay is working. There is no mobility shift detected for the SYP-3 protein in either presence or absence of CIP activity. α-tubulin is used as a loading control. Phos-tag SDS-PAGE was used as for (**D**). (**F**) In vitro kinase assay showing SYP-2 as a potential direct phosphorylation substrate of ERK MPK kinase. Tandem mass (MS/MS) spectrum of the SYP-2 protein subjected to the in vitro kinase reaction. A SYP-2 peptide containing a phosphorylated serine residue (Ser-25), VSFApSPVSSSQK, was identified in this spectrum. The neutral loss from the doubly charged precursor ion is indicative of the phosphorylated peptide. The b and y product ions are indicated in blue and red, respectively. The spectrum is representative of two independent experiments.

The following figure supplement is available for figure 4:

**Figure supplement 1.** SYP-3 does not exhibit a detectable mobility shift in vivo.

region components of the SC, SYP-1, SYP-2, SYP-3 and SYP-4, for the presence of potential MAP kinase phosphorylation sites using the phosphorylation site predictor programs GPS 2.1 and Kinase-Phos 2.0 (*Wong et al., 2007*; *Xue et al., 2008*). This identified potential MAP kinase phosphorylation sites on SYP-1, SYP-2, SYP-4 and none on SYP-3 (*Figure 4A*). Next, to verify the predicted MAP kinase phosphorylation sites in the SYP proteins, we immunoprecipitated the SYP-1/2/3 proteins from lysates of adult worms expressing either SYP-2::GFP or SYP-3::GFP and performed extensive mass spectrometry analysis to look for post-translational modifications (a similar analysis was not possible for SYP-4 given that neither functional antibodies or tagged transgenic lines are currently available). In addition, a phospho-proteomics approach was applied to wild type lysates to further confirm the MAP kinase phosphorylation sites in the SYP proteins (see Materials and methods). These combined approaches confirmed that SYP-2 is phosphorylated at S25, a potential MAP kinase phosphorylation site (*Figure 4A and B*).

We next determined that SYP-2 interacts with the germline-specific isoform of MPK-1 in vivo by immunoprecipitating SYP-2 from the lysates of adult worms expressing endogenous SYP-2 as well as SYP-2::GFP and probing with an MPK-1 antibody on western blots (*Figure 4C*). The reciprocal experiment showed that the SYP-2 protein co-immunoprecipitates with dpMPK-1 in pull downs from adult worm lysates (*Figure 4C*). This in vivo interaction was further confirmed by mass spectrometry analysis where we identified two MPK-1 peptides in the SYP-2 pull down (data not shown).

To further confirm whether SYP-2 is phosphorylated in vivo, we incubated worm lysates with calf intestinal phosphatase (CIP) either in the presence or absence of the phosphatase inhibitor EDTA, and ran these samples on Phos-tag gels for a better separation of phosphorylated from non-phosphorylated bands. CIP treatment results in phosphate removal and a faster running band for SYP-2, whereas when CIP activity is inhibited using EDTA two bands are detected corresponding to the phosphorylated and unphosphorylated forms of SYP-2 (*Figure 4D*). This shows that SYP-2 is phosphorylated in vivo and that SYP-2 exists in both phosphorylated as well as unphosphorylated forms. Further, in *mpk-1(ga117)* null mutants, the upper band corresponding to phosphorylated SYP-2 is not present, suggesting that phosphorylation of SYP-2 is dependent on MAP kinase. In *let-60 (ga89gf)/RAS* gain-of-function mutants, we detect two bands corresponding to the phosphorylated and unphosphorylated SYP-2 protein, instead of only the phosphorylated band. This could be due to only a subset of the SYP-2 protein being phosphorylated by MPK-1. This is consistent with the

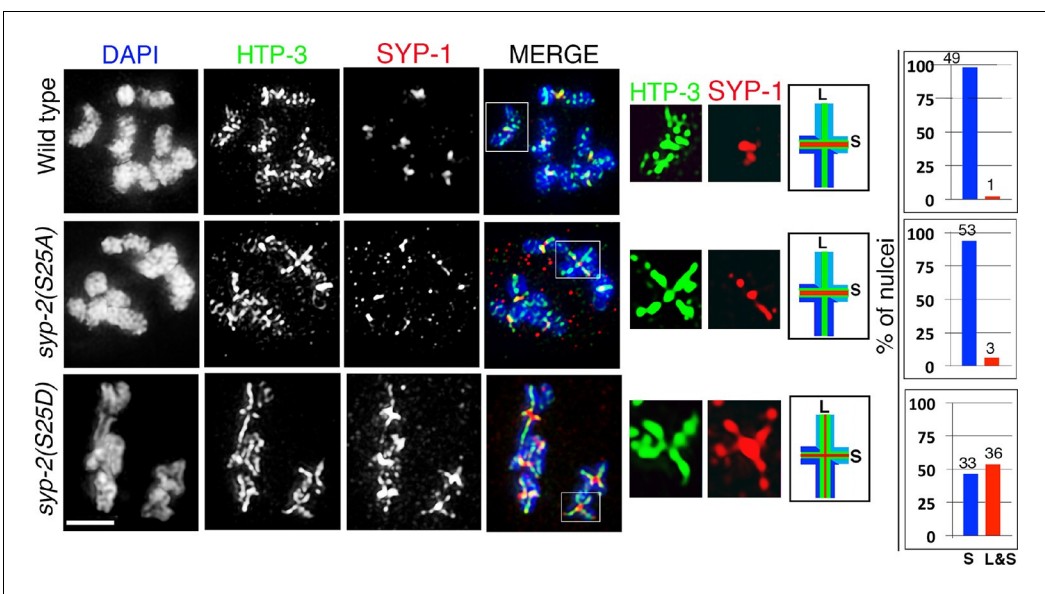

**Figure 5.** Phosphorylation of SYP-2 at S25 is required for normal SC dynamics. High-magnification images of wild type, *syp-2(S25A)* phosphodead, and *syp-2(S25D)* phosphomimetic mutants co-stained with HTP-3 (green), SYP-1 (red) and DAPI (blue). In the *syp-2(S25D)* phosphomimetic mutant the SC fails to disassemble from the long arms of the bivalents in 54% of the nuclei scored. All images are of diakinesis nuclei. White box indicates the bivalent shown at a higher magnification on the right. Bivalents with both long and short arms clearly displayed were chosen for higher magnification. Worms from all the indicated genotypes were grown at 20°C and analyzed 18–24 hr post-L4. Histograms on the right indicate the percentage of diplotene and diakinesis stage nuclei with SYP-1 either only on the short arm (S, blue) or on both long and short arms (red, L&S) of the bivalents. All the bivalents were examined in every nucleus and the bivalents in the same nucleus either all exhibited SYP-1 staining on both the long and short arms or all exhibited staining only on the short arms. Numbers of nuclei scored are shown. n>15 gonads were analyzed for each genotype. Bar, 2 μm.

The following figure supplements are available for figure 5:

**Figure supplement 1.** SYP-1 and HTP-3 localization in *syp-2* phosphodead and phosphomimetic mutants.

**Figure supplement 2.** SYP-2, SYP-3 and SYP-4 localization in *ect-2(gf), let-60(gf)*, and *syp-2* phosphomimetic mutants.

**Figure supplement 3.** SYP-1 and LAB-1 localization in *ect-2(gf), let-60(gf)*, and *syp-2* phosphomimetic mutants.

**Figure supplement 4.** AIR-2 and LAB-1 localization in *syp-2* phosphodead and phosphomimetic mutants.

expression pattern of MPK-1 in the germline where the active form of dpMPK-1 is expressed only from mid to late pachytene. Meanwhile, SYP-2 is expressed from the leptotene/zygotene stage to late diakinesis. Therefore, SYP-2 could be phosphorylated by dpMPK-1 at the mid to late-pachytene region thereby regulating the localization of SYP-2 in a spatio-temporal manner. Further, we found that the SYP-2 (S25A) phosphodead mutant protein runs at the same size of the unphosphorylated SYP-2 band, suggesting that there are no additional phosphorylation sites in the SYP-2 protein (*Figure 4D*). We used SYP-3 as a negative control, since our analyses indicate it lacks phosphorylation sites, and we did not detect any shift in band size (*Figure 4E* and *Figure 4—figure supplement 1*). While this suggests that there may only be an unphosphorylated form of SYP-3, we cannot rule out the possibility that either a small fraction of SYP-3 in the extract tested is phosphorylated or that phosphorylation may not lead to a mobility shift under the conditions utilized in this analysis. We used MPK-1 as a positive control, where the band corresponding to dpMPK-1 (phosphorylated form of MPK-1) is not present after CIP treatment whereas it is present when CIP activity is inhibited

(*Figure 4E*). These data show that SYP-2 is phosphorylated in vivo and that its phosphorylation is dependent on the MAP kinase pathway.

Finally, to determine whether SYP-2 is a potential direct target for phosphorylation by MPK-1, we performed an in vitro kinase assay. Recombinant SYP-2 protein expressed and purified from *E.coli* was incubated with and without the MPK-1 mouse homolog ERK MAP kinase and examined by mass spectrometry to verify phosphorylation. We found that SYP-2 was phosphorylated by MAP kinase at the S25 site in vitro in the presence of ERK MAP kinase (*Figure 4F*). The phosphorylated peptide was not observed in the absence of the kinase or when the kinase reaction was subsequently incubated with lambda phosphatase. Altogether, our data indicate that in vivo phosphorylation of SYP-2 is dependent on the MAP kinase pathway and that SYP-2 is an in vitro phosphorylation substrate of the ERK MAP kinase.

## Phosphorylation of SYP-2 at S25 prevents disassembly of SC proteins from the long arms of the bivalents

To determine the role for the MPK-1-mediated phosphorylation of SYP-2, we generated phospho-dead and phosphomimetic SYP-2 mutants using CRISPR-Cas9 technology (*Tzur et al., 2013*). We generated lines where the S25 residue was mutated to either an alanine to generate a phosphodead mutant or to aspartic acid to generate a phosphomimetic mutant. We did not observe defects in either SC assembly or disassembly in the *syp-2* phosphodead mutant (*Figure 5* and *Figure 5—figure supplement 1*). This result was not surprising since the *mpk-1(ga111lf)* mutant did not show any defects in the disassembly of SC proteins at the non-permissive temperature and we already hypothesized that the SC assembly defects we see in both *mpk-1* loss- and gain-of-function mutants could be due to other roles played by MPK-1 in germline development or through its effects on different target proteins. However, in the *syp-2* phosphomimetic mutant, while SC assembly was normal, the disassembly of SC proteins was impaired as exemplified by SYP-1, SYP-2, SYP-3 and SYP-4 persisting on the long arms and failing to become restricted to the short arms of the bivalents (*Figure 5*, *Figure 5—figure supplement 2* and *Figure 5—figure supplement 3*). This is consistent with the observation that constitutive activation of dpMPK-1 leads to failure in disassembly of SC proteins from the long arm of the bivalents (this study), and that all four SYP proteins are interdependent on each other for their localization (*Colaiácovo et al., 2003*; *Smolikov et al., 2007a*; *2009*). Other aspects of late prophase chromosome remodeling, such as AIR-2 and LAB-1 restricted localizations, are not altered in either the *syp-2* phosphodead or phosphomimetic mutants (*Figure 5—figure supplement 3* and *4*). However, we see a reduction in brood size (109 ± 32), accompanied by 14.3% embryonic lethality and 2.1% males, in *syp-2* phosphomimetic mutants indicating that impaired disassembly of the SC proteins results in defects in chromosome segregation. Taken together, our data show that phosphorylation of SYP-2 protein at the S25 site prevents the disassembly of the SC proteins from the long arm of the bivalents during late prophase.

## MPK-1 coordinates meiotic recombination with disassembly of SC proteins

Disassembly of SC proteins is impaired and the SYP proteins fail to be properly restricted to a chromosome subdomain during late prophase in crossover-defective mutants such as *spo-11, msh-5, msh-4, zhp-3* and *cosa-1*, which instead often continue to retain the SYP proteins along the full lengths of the chromosome axes (*Bhalla et al., 2008*; *Nabeshima et al., 2005*; *Yokoo et al., 2012*; *Martinez-Perez et al., 2008*). Our study of HIM-18/Slx4, which is involved in the resolution of Holliday junction intermediates into COs, indicated that the symmetry-breaking event detected at late pachytene is likely a CO precursor and not a completed CO event (*Saito et al., 2009*). Specifically, analysis of *him-18* null mutants showed six ZHP-3 foci per oocyte marking the six CO precursor sites also observed in wild type during late pachytene, and the establishment of long and short arm domains as indicated by LAB-1, SYP-1 and AIR-2 localization. Since our current study indicates that the MPK-1 pathway regulates the disassembly of SC proteins we tested the hypothesis that MPK-1 coordinates recognition of a CO precursor/intermediate with disassembly of SC proteins from the long arms of the bivalents. First, active dpMPK-1 and foci for the pro-CO factor COSA-1 are both observed at the same time in pachytene stage nuclei suggesting a correlation between CO precursors and active MPK-1 (*Figure 6A*). Second, we observed a significant persistence of dpMPK-1 signal

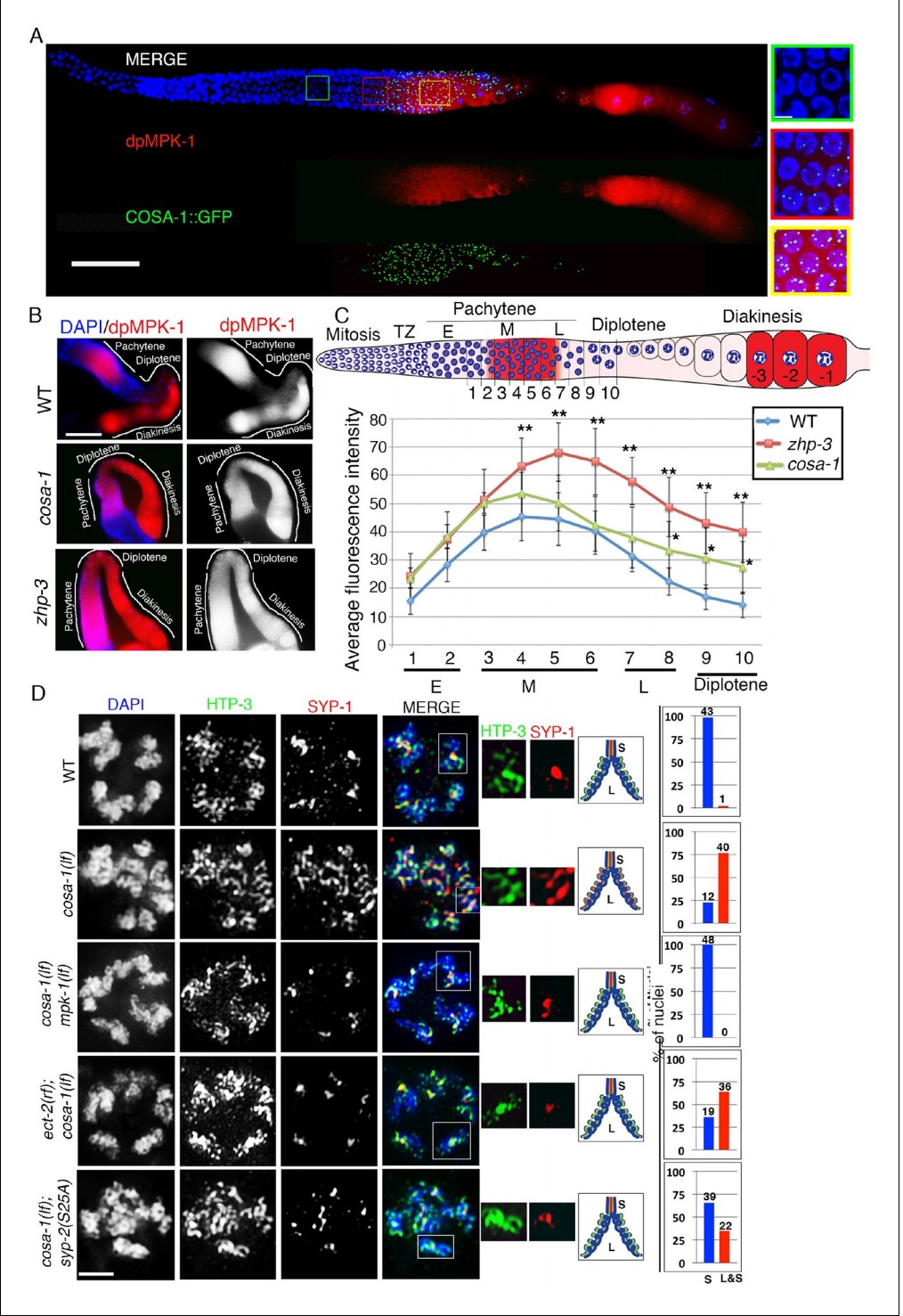

**Figure 6.** MPK-1 links CO designation with the disassembly of SC proteins. (**A**) Low-magnification images of whole mounted gonads depicting COSA-1::GFP and dpMPK-1 localization in wild type. Signals for the pro-crossover marker COSA-1::GFP and dpMPK-1 are both observed at the mid to-late pachytene region. Insets show higher magnification images of different regions in the germline: (green inset) Early to mid-pachytene, no COSA-1 localization and dpMPK-1 is off; (red inset) mid to late-pachytene, COSA-1::GFP starts to localize on the chromosomes concomitant with the appearance of the dpMPK-1 signal; (yellow inset) late pachytene, 6 COSA-1:: GFP foci/nucleus are observed and strong dpMPK-1 signal is detected. 20 gonads were analyzed. Bars, 20μm for whole gonad and 2μm for insets. (**B**) Low-magnification images of wild type, *cosa-1* and *zhp-3* gonads co-stained

*Figure 6 continued on next page*

*Figure 6 continued*

with dpMPK-1 (red), and DAPI (blue). dpMPK-1 expression level is not turned off in the *cosa-1* and *zhp-3* mutants in late pachytene and diplotene in contrast to wild type. n>18 gonads were analyzed for each. Bar, 20 µm. (**C**) Graph showing the quantification of dpMPK-1 fluorescence signal intensity in different regions of the gonad arm in wild type, *zhp-3* and *cosa-1* mutants. **p<0.005, *p<0.02 (unpaired student's t-test; n>18, 17 and 15 gonad arm were analyzed for wild type, *zhp-3* and *cosa-1* mutants, respectively). Regions where dpMPK-1 signal was quantified are indicated on the diagram depicting the hermaphrodite germline. TZ stands for transition zone (leptotene/zygotene). E, M and L stand for early, mid and late pachytene, respectively. Progression from mitosis into meiosis is displayed from left to right and the last three oocytes at diakinesis (-3 to -1) are indicated. (**D**) Co-staining with HTP-3 (green), SYP-1 (red) and DAPI (blue) of diplotene nuclei from the indicated genotypes. Illustrations depict the bivalent configuration at this stage. S indicates short arm and L indicates long arm. White boxes indicate the bivalent shown at a higher magnification on the right. Bivalents with both long and short arms clearly displayed were chosen for higher magnification. Histograms on the right indicate the percentage of diplotene and diakinesis stage nuclei with SYP-1 either only on the short arm (S, blue) or on both long and short arms (red, L&S) of the bivalents. All the bivalents were examined in every nucleus and the bivalents in the same nucleus either all exhibited SYP-1 staining on both the long and short arms or all exhibited staining only on the short arms. Numbers of nuclei scored are shown. Bar, 2 µm. Worms from all the genotypes indicated in (**A–C**) were grown at 20°C and analyzed 18–24 hr post-L4. Worms from all the genotypes indicated in (**D**) were grown at 15°C, shifted to 25°C at the L4 stage, and analyzed 18–24 hr post-L4.

in *zhp-3* and *cosa-1* mutants compared to wild type suggesting that inactivation of MPK-1 is linked to formation of a CO precursor (*Figure 6B and C*). Third, the defect in disassembly of SC proteins observed in *cosa-1* mutants is fully rescued in a *cosa-1 mpk-1(lf)* double mutant and partly rescued in *ect-2(rf);cosa-1* and *cosa-1;syp-2(S25A)* double mutants (11.5% and 40.9% suppression of the *cosa-1* disassembly defect phenotype, respectively) (*Figure 6D*). The partial suppression observed in *ect-2(rf);cosa-1* mutants can be ascribed to the fact that *ect-2(rf)* is a reduction-of-function mutant and still has dpMPK-1 activity as indicated in *Figure 2—figure supplement 5* as well as by the presence of phosphorylated SYP-2 detected on Westerns (*Figure 4D*). The *cosa-1; syp-2(S25A)* partial rescue leads us to hypothesize that there may be additional kinases regulating the other SYP proteins in an MPK-1-dependent manner. Indeed, while we did not find evidence for additional MPK-1 phosphorylation sites on SYP-2, there are additional phosphorylation sites for other kinases, such as the polo-like kinase, on SYP-1 and SYP-4, verified by mass spectrometry (Nadarajan and Colaiacovo, unpublished and PHOSIDA (http://www.phosida.com) and Polo-like kinase itself was identified as a target of MPK-1 in *C. elegans* (*Arur et al., 2009*). While the roles exerted by these additional kinases on the SC remain to be investigated, our observations support a role for MPK-1 in coordinating recognition of a CO precursor with asymmetric disassembly of SC proteins.

## Discussion

Phosphorylation is one of the key post-translational modifications playing a crucial role in regulating fundamental cellular processes from yeast to humans. Here we provide novel insight into the regulation of the disassembly of SC proteins from along the long arms of the bivalents in response to progression through the CO specification pathway via regulation of phosphorylation of a central region component of the SC. First, we discovered a novel role for ECT-2 in the regulation of SC dynamics during prophase I of meiosis. Second, we showed that ECT-2 functions through the conserved MAP kinase pathway to regulate the disassembly of SC proteins. Third, we showed that SYP-2 is a potential direct target of phosphorylation by the MAP kinase pathway both in vivo and in vitro, and that constitutively phosphorylated SYP-2 impairs the disassembly of SC proteins from the long arms of the bivalents. Fourth, we showed that pro-crossover factor COSA-1 and dpMPK-1 are expressed at the same time in the pachytene region, and that inactivation of dpMPK-1 at the late pachytene and diplotene regions requires the pro-crossover activities of ZHP-3 and COSA-1. Taken together, these data identify the MAP kinase-dependent phosphorylation of SYP-2 as part of a mechanism set in place to trigger the disassembly of SC proteins once CO designation occurs.

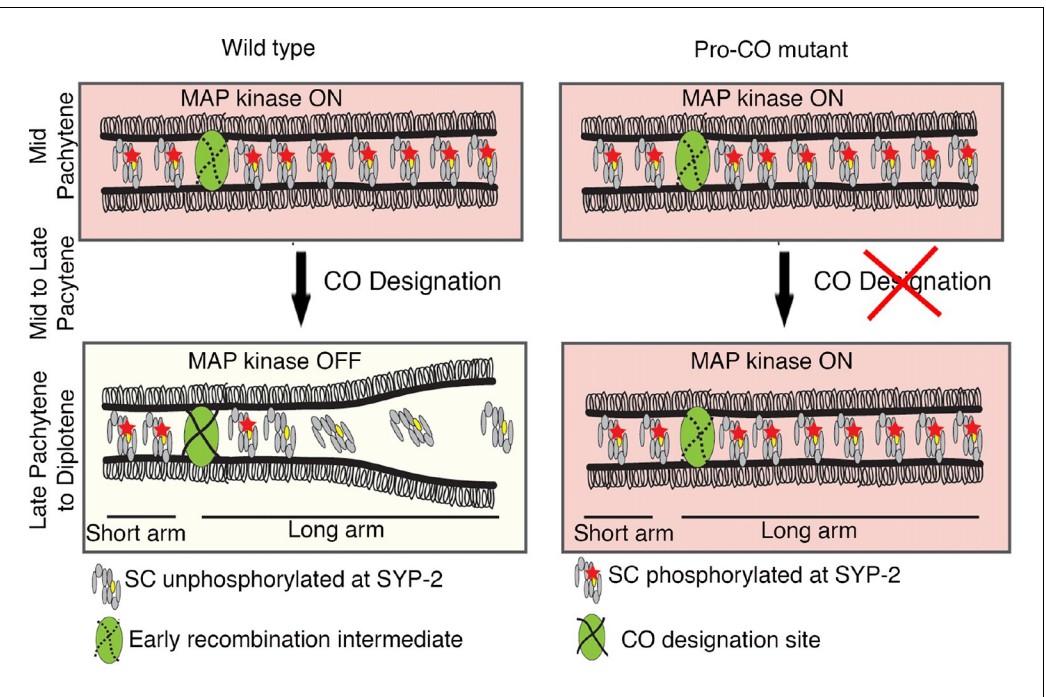

**Figure 7.** Model for MPK-1-mediated coordination between CO designation and the disassembly of SC proteins during meiosis. Proposed model for MPK-1 function in coordinating CO designation and the disassembly of SC proteins from the long arms of the bivalents. In mid-pachytene, dpMPK-1 phosphorylates the SYP-2 protein at the S25 site and CO specification is progressing. When CO designation is detected, MAP kinase is turned off, SYP-2 is either dephosphorylated along the long arms of the bivalents by a yet unknown phosphatase or, given the dynamic nature of the SC, phosphorylated SYP-2 is replaced by unphosphorylated SYP-2 in this region and the SC starts to disassemble from the long arms in late pachytene/diplotene. In pro-CO mutants, such as *cosa-1* and *zhp-3*, CO designation is impaired and dpMPK-1 persists at the late pachytene and diplotene regions preventing disassembly of the SC proteins.

The following figure supplement is available for figure 7:

**Figure supplement 1.** Conserved MAP kinase phosphorylation sites present in central region components of the mammalian SC.

## An MPK-1 mediated surveillance mechanism links CO designation to disassembly of SC proteins from the long arms of the bivalent

The central region component of the SC in budding yeast, Zip1, is continuously added to the SC during meiotic prophase and exhibits differential dynamics near the recombination site (*Voelkel-Meiman et al., 2012*). Similarly, the *C. elegans* SC has been proposed to be a dynamic structure that can disassemble and reassemble locally in response to exogenous DNA damage induced by irradiation (*Couteau and Zetka, 2011*). Here, we identify a regulatory mechanism underlying the important step of disassembly of the SC proteins from the long arms of the bivalents. Our studies have revealed the existence of both a phosphorylated and an unphosphorylated pool of SYP-2 protein. We propose that SYP-2 is phosphorylated by MPK-1 at the S25 site and that this must take place during the mid to late pachytene stage when activated MPK-1 is expressed. Our studies revealed that a constitutively phosphorylated SYP-2 resulted in an inability to remove the SYP proteins from the long arms of the bivalents in a timely manner. However, an inability to phosphorylate SYP-2 at S25 does not affect either SC assembly or disassembly, and we did not detect residual or novel phosphorylation in *syp-2(S25A)* mutants in our CIP assays. These observations lead us to suggest that phosphorylation of SYP-2 at S25 may 'prime' this protein for subsequent dephosphorylation once crossover designation is detected. This would ensure that homologous interactions are not prematurely dissolved until CO specification takes place, thus guaranteeing that homologs are

locked and ensuring subsequent accurate chromosome segregation. The CO precursor markers and dpMPK-1 are expressed during the same time in the germline (*Figure 6A*), suggesting that the timing when CO designation is completed coincides with when dpMPK-1 is turned off (germ cells progress from pachytene to diplotene). Our model is that MPK-1 functions as part of a surveillance mechanism in the pachytene region that recognizes the CO precursor and triggers the disassembly of SC proteins from along the long arms of the bivalents upon CO designation (*Figure 7*). Our model is further supported by the impaired disassembly of SC proteins detected in the absence of CO designation in *zhp-3* and *cosa-1* mutants (*Bhalla et al., 2008*; *Yokoo et al., 2012*) and our observation that dpMPK-1 persists into late pachytene and diplotene in *zhp-3* and *cosa-1* mutants (*Figure 6B and C*).

## The disassembly of SC proteins occurs in two independently regulated steps

The SYP proteins are removed from the chromosomes in two steps, first SYPs are lost from the long arms of the bivalents in the late pachytene to early diplotene stage nuclei, but are selectively maintained on the short arms of the bivalents until late diakinesis (-3 oocyte; *Figure 6C*). In the second step, the SYP proteins are lost from the short arms of the bivalents. This second round of removal might be regulated in a manner independent of the MAP kinase pathway. This is supported by our observation that the SYP proteins are able to disassemble from the short arms of the bivalents in *fog-2(oz40)* female animals where MPK-1 is not active in the proximal oocytes, in a manner indistinguishable from wild type (Nadarajan and Colaiacovo, unpublished results). How then are the SYP proteins selectively lost from the long arms but not from the short arms of the bivalents during the diplotene stage? There are several possible ways this could be carried out: 1) There could be proteins localizing to the short arms of the bivalents in late pachytene to prevent loss of the SYP proteins specifically from the short arms; 2) a phosphatase could specifically localize to the long arms of the bivalents and dephosphorylate the SYP-2 protein located only on that chromosome subdomain thereby leading to loss of SYP proteins from the long arms while localization of this phosphatase to the short arms of the bivalents could be blocked by other proteins; and 3) there might be factors preventing the addition of unphosphorylated SYP-2 protein on the short arms of the bivalents. In fact, there are precedents for such modes of regulation acting on the short/long arms of the bivalents, as exemplified by LAB-1, the ortholog of Shughoshin in *C. elegans*, and the HORMA domain containing proteins HTP-1/HTP-2. LAB-1 localizes exclusively to the long arm of the bivalents in late prophase where it prevents Aurora B Kinase, AIR-2, from loading on the long arm of the bivalents and restricts its localization to the short arm by recruiting the PP1/Glc7 phosphatases GSP-1 and GSP-2, thereby ensuring homologous chromosome segregation during meiosis I and sister chromosome segregation during meiosis II (*de Carvalho et al., 2008*; *Tzur et al., 2012*). Similar to LAB-1, the HORMA domain containing HTP-1/2 proteins relocalize from being associated along the full length of the chromosomes to being exclusively localized to the long arm of the bivalents during chromosome remodeling in late prophase I (*Martinez-Perez et al., 2008*). It has been suggested that differentially regulated pools of HTP-1/2 near the chromosome axes and/or post-translational modifications such as phosphorylation within the closure motifs of HTP-3, which remains associated with both long and short arms and with whom HTP-1/2 interact, could contribute to their distinct localization (*Kim et al., 2014*).

## Regulation of asymmetric disassembly of SC proteins – an evolutionarily conserved mechanism?

How conserved is the asymmetric disassembly of SC proteins? This process occurs not only in *C. elegans,* but also in budding yeast, flies and mice. In both yeast and flies, the SC disassembles asymmetrically and persists associated with the centromere until late prophase (*Newnham et al., 2010*; *Takeo et al., 2011*; *Gladstone et al., 2009*). Similarly in mouse spermatocytes, the SC is lost along the length of the chromosome but is retained at paired centromeres (*Qiao et al., 2012*; *Bisig et al., 2012*). Studies in yeast show that the SC that persists at centromeres, after the SC disassembles throughout other regions of the chromosomes, promotes centromere bi-orientation thereby ensuring homologous chromosome segregation at meiosis I (*Gladstone et al., 2009*). In mouse spermatocytes, the SC that persists between the chiasmata, after SC disassembly in the pachytene region, is

proposed to regulate local remodeling of homologous chromosome axes thereby promoting centromere and chiasmata functions, which are to ensure proper homolog segregation at meiosis I (*Qiao et al., 2012*). While chromosomes in *C. elegans* are holocentric, and therefore lack a localized centromere, we propose that the CO designation-triggered asymmetric disassembly of SC proteins, and concomitant chromosome remodeling, are set in place to ensure that the short arm of the bivalents can function in a manner orthologous to centromeres to promote bi-orientation of the homologs thereby ensuring accurate segregation of homologous chromosomes during meiosis I suggesting the existence of a conserved regulatory mechanism. Further support for this stems from the demonstration that chromosome segregation at meiosis I in *C. elegans* occurs by outward pushing forces from microtubules assembling at the interface between the short arms of the separating homologs (*Dumont et al., 2012*). Lastly, both mammalian ECT2 and the ERK/MAP kinase are expressed in testis and ovaries (*Hirata et al., 2009*; *Fields and Justilien, 2010*; *Inselman and Handel, 2004*; *Nissan et al., 2013*; *Uhlén et al., 2015*). We have found that some of the central region components of the mammalian SC, namely SYCE1, SYCE2 and TEX12, contain putative MAP kinase phosphorylation sites predicted by GSP 2.0 ([*Wong et al., 2007*; *Xue et al., 2008*]; *Figure 7—figure supplement 1*). Moreover, the predicted ERK/MPK kinase phosphorylation sites in SYCE1 (S308) and TEX12 (S27) are conserved between mice and humans, and the S308 site in SYCE1 has been confirmed by mass spectrometry analysis as being phosphorylated in vivo in mice (*Huttlin et al., 2010*), raising the interesting possibility that we uncovered a conserved mode of regulation for asymmetric disassembly of SC proteins shared between worms and mammals.

## Materials and methods

### *C. elegans* strains and genetics

The *C. elegans* N2 Bristol strain was used as the wild-type background and worms were cultured under standard conditions as described in (*Brenner, 1974*). Temperature sensitive strains (ts) were grown at the permissive temperature of 15°C and transferred to the restrictive condition of 25°C at the L4 stage. 24 hr post-L4 worms were then analyzed for the mutant phenotypes. The wild type worms used for comparisons with these mutants were all subjected to the same temperature shifts and examined at the same times as the mutants. The following mutations and chromosome rearrangements were used: LG II: *ect-2(e1778)/dyp-10(e128) II, ect-2(ax751) II, unc-4(e120) ect-2(zh8) II/ mln1 [dpy-10(e128) mls14], unc-4(e120) II, ect-2(gk44) II; unc-119(ed3) III; xnls162 [ect-2::GFP + unc-119(+)], ect-2(ax751) II; unc-119(ed3) III; xnls162 [ect-2::GFP + unc-119(+)], ect-2(zh8) II/mln1 [dpy-10 (e128) mls14]; mpk-1(ga111) III, ect-2(ax751) II; let-60(ga89) IV; ect-2(ax751) II; cosa-1(me13) III;* LG III: *mpk-1(ga117)/dpy-17(e164) unc-79(e1068) III, mpk-1(ku1) unc-32(e189) III, unc-32(e189) III, mpk-1 (ga111) III, mels9(unc-119(+) pie-1promoter::gfp::syp-3); unc-119(ed3) III; cosa-1(me13) mpk-1 (ga111) III; cosa-1(me13)III; syp-2(rj16(S25A) IV;* LG IV: *lip-1(zh15) IV, let-60(ga89) IV;* and LG V: *syp-2 (ok307) V/nT1 [unc-?(n754) let-?(m435)] (IV;V), wgls227 [syp-2::TY1::EGFP::3xFLAG(92C12) + unc-119, syp-2(rj16(S25A) V, and SYP-2(rj17(S25D).*

*ect-2* was identified in a targeted RNAi screen on 168 germline-enriched genes designed to identify novel components regulating chromosome remodeling and short/long arm identity on bivalents. Specifically, we utilized a GFP-tagged AIR-2 containing line (*ojls50 (pie-1p::GFP::AIR-2 + unc-119 (+)*) to screen for candidates that when depleted resulted in the mislocalization of the Aurora B kinase, AIR-2, which localizes to the short arms of the bivalents during late prophase I of meiosis to ensure accurate chromosome segregation. Criteria for the selection of these genes are detailed in (*Colaiácovo et al., 2002*) and were applied to germline-enriched genes identified by microarray analysis in (*Reinke et al., 2004*).

*ect-2(ax751rf)* mutants phenocopied the *ect-2 (RNAi)* phenotype at the non-permissive temperature with AIR-2 failing to load onto the chromosomes (*Figure 1A*). However, unlike *ect-2(RNAi)*, where 100% of animals (N=44/44) showed failure in AIR-2 loading on the chromosomes, only 52% of the *ect-2(ax751rf)* mutants (N=13/25) exhibited this defect, which could be due to the fact that *ect-2 (ax751)* is not a null mutant.

Since the *ect-2(zh8gf)* and the *mpk-1(ku1)* mutants are in the *unc-4* and *unc-32* mutant backgrounds, respectively, we analyzed SC dynamics in *unc-4* and *unc-32* mutants. We did not find any defects in either SC assembly or disassembly in *unc-4* and *unc-32* mutants (*Figure 2—figure*

supplement 2). Analysis of *mpk-1(ga117)* null mutants revealed defects in SC assembly, indicated by the formation of polycomplexes at 20°C (15%, n=20, where n is the number of gonads scored) and 25°C (66%, n=16).

## Antibodies

Primary antibodies were used at the following dilutions for immunofluorescence: chicken α-GFP (1:400; Abcam, Cambridge, MA), rabbit α-SYP-1 (1:200;(*MacQueen et al., 2002*), guinea pig α-HTP-3 (1:400; (*Goodyer et al., 2008*), rabbit α-AIR-2 (1:100;[*de Carvalho et al., 2008*]), rabbit α-LAB-1 (1:300;[*de Carvalho et al., 2008*]), mouse α-dpMPK-1 (1:500; Sigma, St. Louis, MO) and rabbit anti-RhoA (1:200; Santa Cruz). The following secondary antibodies from Jackson ImmunoResearch (Jackson ImmunoResearch, WestGrove, PA) were used at a 1:200 dilution: α-chicken FITC, α-rabbit Cy5, α-mouse FITC, and α-guinea pig FITC. Vectashield containing 1μg/μl of DAPI from Vector Laboratories was used as a mounting media and anti-fading agent.

Primary antibodies were used in the following dilutions for western blot analysis: chicken α-GFP (1:2000; Abcam), rabbit α-SYP-2 (1:200;[*Colaiácovo et al., 2003*]), rabbit α-SYP-3 (1:200; [*Smolikov et al., 2007b*]), mouse α-dpMPK-1 (1:500; Sigma), mouse α-tubulin (1:2000; Sigma) and rabbit α-MPK-1 (1:2000; Santa Cruz, Dallas, TX). HRP-conjugated secondary antibodies, donkey anti-chicken, rabbit anti-mouse, and mouse anti-rabbit from Jackson ImmunoResearch were used at a 1:10,000 dilution.

## Immunofluorescence and imaging

Whole mount preparation of dissected gonads and immunostaining procedures were performed as in *Colaiácovo et al. (2003)*. Immunofluorescence images were captured with an IX-70 microscope (Olympus, Waltham, MA) fitted with a cooled CCD camera (CH350; Roper Scientific, Tuscon, AZ) driven by the Delta Vision system (Applied Precision, Pittsburgh, PA). Images were deconvolved using the SoftWorx 3.0 deconvolution software from Applied Precision. Flourescent images showing RHO-1 expression and dpMPK-1 expression were captured with a Zeiss Axioskope microscope fitted with a Hamamatsu digital CCD camera.

## In vitro kinase assay

Mouse recombinant p42 MAPK (Erk2) was purchased from New England Biosciences (Cat #-P6080S; Ipswich, MA). Full-length recombinant SYP-2, expressed and purified from *E. coli*, was used for in vitro kinase assays. *syp-2* cDNA from *C. elegans* was cloned into the pET30a vector and expressed and purified from *E. coli* following the protocol published in *Jambhekar et al. (2014)*. Kinase reactions were performed as described previously (*Arur et al., 2011*). Briefly, 20 μM SYP-2 was incubated with 20 U p42 MAPK in the presence of 250 μM ATP and 1X kinase buffer (NEB, Cat #- B6022S) for 30 min at 30°C. In the control reaction, p42 MAPK was not included. Reactions were performed in duplicate. For the phosphatase treated reaction, the kinase reaction was carried out as described above. Following completion of the reaction, the kinase activity was inhibited with 400 μM ERK Inhibitor II, FR180204 (Santa Cruz, Cat. #- sc-203945). 100 U lambda phosphatase (NEB, Cat #- P0753S), supplemented with 1 mM MnCl2 was then added and the sample incubated for an additional 30 min at 30°C. All reactions were terminated by adding 2X SDS sample buffer and boiling for 3 min. The samples were resolved by SDS-PAGE electrophoresis, stained with Coomassie blue, the bands corresponding to the molecular weight of SYP-2 excised, and subjected to in-gel digestion with Trypsin as previously described (*Shevchenko et al., 2006*). Following overnight digestion at 37°C, peptides were extracted with 5% formic acid/50% acetonitrile, purified over Empore C18 extraction media (3M), and analyzed by liquid chromatography- tandem mass spectrometry (LC-MS/MS) with a LTQ-Velos linear ion trap mass spectrometer (Thermo Scientific, Cambridge, Mass) with an 18 cm3 125 μm (ID) C18 column and a 50 min 10–35% acetonitrile gradient. MS/MS spectra searches were performed using Sequest (*Eng et al., 1994*).

## Generation of mutants via the CRISPR-Cas9 system

The CRISPR-Cas9 genome editing technology was used to engineer *syp-2* phosphodead and phosphomimetic mutations at the endogenous locus (*Tzur et al., 2013*). To generate a phosphodead *syp-2* mutant, serine 25 (S25A) was mutated to alanine. To generate a phosphomimetic mutant,

serine 25 (S25D) was mutated to aspartic acid. We used the sgRNA recognition site (gaaaacagctg-cagtaactgtgg) 139 base pairs downstream of the start codon. To express the sgRNA, we replaced the *unc-119* recognition sequence (gaattttctgaaattaaaga) in the *pU6::unc-119*_sgRNA plasmid (*Friedland et al., 2013*) with the SYP-2 sequence (gaaaacagctgcagtaactg). The donor sequence containing the genomic sequence of *syp-2* extending from 1380 bp upstream to 1821 bp downstream of the start codon with the following changes: TC to GA change at position 1192 to generate the phosphomimetic mutant and a T to G change at position 1192 to generate the phosphodead mutant and a G to C change at position 160 (resulting in a silent mutation that is expected to prevent re-cutting by the Cas9), was cloned into the BglII site of the pCFJ104 vector expressing *Pmyo-3::mCherry:: unc-54* (*Frøkjaer-Jensen et al., 2008*). A cocktail consisting of a plasmid expressing the sgRNA (200 ng/µl), a plasmid expressing the donor sequence (97.5 ng/µl), Cas9 (200 ng/µl) and the co-injection marker pCFJ 90 (*Pmyo-2::mCherry::unc-54utr*; 2.5 ng/µl) was microinjected into the gonad arms of the worms (P0s). F1 animals expressing the co-injection marker were sequenced to identify mutants and homozygous animals were picked from among the F2 generation and confirmed by sequencing.

## Immunoprecipitation

Wild type animals expressing SYP-2::GFP and GFP::SYP-3 were grown in large quantity using 8-times peptone-enriched plates seeded with NA22 bacteria. 24 hr post-L4 worms were collected and washed 3 times with M9 and then one time with lysis buffer (50 mM Hepes buffer, pH 7.5, 1 mM MgCl2, 300 mM KCl, 10% glycerol, 0.05% NP-40, 5 mM 2-me and protease inhibitor [protease inhibitor cocktail tablet, catalog # 11836153001; from Roche, Branford, CT]). Worms were frozen in liquid nitrogen followed by grinding using mortar and pestle. Frozen ground worms were mixed with an equal volume of lysis buffer and sonicated in a Diagenode Bioruptor sonication water bath (Bioruptor Sonication, VCD300) alternating for 15 s on and 45 s off for a total of 20 min. Crude worm lysate was spun down at 14000 rpm and filtered with a 0.45um syringe filter (Millex-HP filter unit, Catalog # SLHP033RS). 500 µl of worm lysate was incubated with 30 µl anti-GFP conjugated agarose beads (MBL, Catalog # D153-8) overnight at 4°C on a nutator. Beads were washed 3 times with 1XPBS and boiled in 2xSample buffer for 5 min. For SYP-2 interaction with dpMPK-1, supernatant was run on a 4–15% gradient gel and protein-protein interaction was verified on western blots with the appropriate antibody. Anti-IgG conjugated agarose beads were used as control. To verify the interaction by mass spectrometry, protein in the supernatant was precipitated using the proteoExtract protein precipitation kit (CALBIOCHEM, catalog # 539180) followed by mass spectrometry analysis. To identify phosphorylation sites on the SYP proteins, the supernatants were run on 4–15% gradient gel, stained with a Pierce silver stain kit (catalog # 24612) and bands of the appropriate size were analyzed by mass spectrometry.

## Calf intestinal phosphatase assay

Immunoprecipitation was prepared as mentioned above. Thirty microliters of immunoprecipitate were incubated with 20 units of calf intestinal phosphatase (NEB catalog # M0290S) for 1 hr at 37°C. 50 mM EDTA was used as phosphatase inhibitor. The immunoprecipitate was purified after CIP treatment by using a Pierce SDS-PAGE sample prep kit (catalog # 89888). The sample was boiled for 5 min and run on a SuperSep phos-tag 12.5% precast gel (catalog # 195–16391 from Wako) to get better separation of phosphorylated from non-phosphorylated protein. To improve the efficiency of protein transfer to PVDF membrane, gel was soaked in transfer buffer with 5 mmol/L EDTA for 10 min with gentle agitation then washed with transfer buffer for 10 min. Western blot was developed with Pierce ECL plus Western blotting substrate (Catalog number # 32132).

## RNA interference

RNAi was performed as in *Govindan et al. (2006)* with the following modifications: three L4-stage animal were placed on each RNAi plate and next generation 24 hr post-L4 animals were screened for phenotype. HT115 bacteria expressing empty pL4440 vector was used as the control RNAi.

Strong RNAi knockdown of *rho-1* leads to severe cytokinesis defects, large nuclei in pachytene, a disrupted plasma membrane organization, a small oogenic germline and an abnormal oocyte progression. Strong RNAi knockdown of *rho-1* also leads to little or no activated MPK-1 (dpMPK-1)

staining. However, it is possible that severe reduction of dpMPK-1 staining is an indirect effect of highly abnormal germline following strong loss of *rho-1* function. Therefore, partial RNAi knockdown of *rho-1* was performed resulting in germlines with wild type morphology/organization. Partial *rho-1* depletion was achieved by feeding RNAi started at the mid/late-L4 stage for 24 hrs at 20°C followed by gonad dissection and staining.

## Phosphorylation site identification
Wild-type worms were grown and nuclei isolated as described in (*Silva et al., 2014*). Subsequent phosphopeptide preparation and identification are described in detail in Supplemental Information.

## Supplemental materials and methods
### Phosphorylation site identification
Wild-type worms were grown and nuclei isolated as described in (*Silva et al., 2014*). Frozen nuclear pellets were solubilized in 250 µL of 8 M urea, 20 mM HEPES containing phosphatase inhibitors (sodium orthovanadate 100 mM, sodium fluoride 500 mM, glycerol phosphate 1 M, disodium pyrophosphate 250 mM). Samples were thawed at room temperature on an Eppendorf mixer. Protein amount was normalized to 500 µg and made up to 1 mL in 8 M urea, 20 mM HEPES. Sample disulfide bridges were reduced by addition of 10 mM dithiothreitol and incubated for 15 min at room temperature in the dark with mixing. Reduced cysteine residues were capped by addition of 25 mM iodoacetamide and incubated for 15 min at room temperature in the dark. Addition of 3 mL of 20 mM HEPES reduced the urea concentration to below 2 M, thereby permitting trypsin digestion. 320 µL of immobilised TPCK trypsin bead slurry (P/N 20230 Thermo Scientific) was washed three times with 20 mM HEPES to remove storage buffers then split evenly between the two samples. Samples were incubated for 18 hr at 37°C with mixing. Digestion was halted by acidifying the samples with <pH 4 by addition of 1% trifluoroacetic acid (TFA). Samples were centrifuged to pellet the trypsin beads and the supernatant, containing peptides, was recovered. Peptides were desalted by reversed-phase chromatographic cartridges (Oasis HLB, Waters Corporation). Cartridges were activated in neat acetonitrile (MeCN) then washed twice with 2% MeCN containing 0.1% TFA. Samples were loaded and permitted to bind to the column slowly under vacuum. Bound peptides were washed twice with 2% MeCN containing 0.1% TFA and then eluted with 1 mL of 1 M glycolic acid in 50% MeCN, 5% TFA. Phosphopeptides were enriched from the eluted peptide solution through use of titania beads (TiO2, Titansphere, 5 µm, GL Sciences Inc.) as previously described (*Casado et al., 2014*). Briefly, 50 µL of titanium dioxide beads in 1% TFA was added to each sample then incubated for 5 min with agitation. Glygen filter-tips were washed with neat MeCN then the sample was loaded into the tips. The tips were centrifuged and flow-through was discarded leaving the titanium beads with bound peptides above the filter. Samples were washed with 1 M glycolic acid in 80% MeCN, 5% TFA to remove non-phosphorylated peptides (discarded) and then washed with 100 mM ammonium acetate in 25% MeCN to remove acidic non-phosphorylated peptides (discarded). Any remaining salts or non-phosphorylated peptides were removed by washing with 10% MeCN in triplicate. Phosphorylated peptides were eluted by four successive 50 µL volumes of 10% MeCN, 5% ammonium hydroxide solution into a 1.5 mL Protein Lo-bind Eppendorf. Eluted phosphopeptides were dried under vacuum in a speed vac concentrator. Dried phosphopeptides were solubilized in 25 µL of 0.1% TFA and transferred to a hydrophobic insert tube within an autosampler vial. Both samples were loaded into a temperature controlled autosampler within an UltiMate 3000 RSLC nanoLC instrument coupled online to an LTQ-Orbitrap Velos mass spectrometer (Thermo Scientific). Sample loading was achieved with a flow of 8 µL/min onto a trap column (Thermo Scientific Acclaim Pepmap 100; 100 µm internal diameter, 2 cm length, C18 reversed-phase material, 5 µm diameter beads, 100Å pore size) in 98% water, 2% MeCN, 0.1% TFA. Peptides were then eluted on-line to an analytical column (Thermo Scientific Acclaim Pepmap RSLC; 75 µm internal diameter, 25 cm length, C18 reversed-phase material, 2 µm diameter beads, 100 Å pore size) and separated using a gradient with conditions: initial 5 min with 4% B (96% A), then 120 min gradient 4-45% B, then 10 min isocratic at 100% B, then 5 min isocratic at 4% B (solvent A: 98% water, 2% MeCN, 0.1% formic acid; solvent B: 20% water, 80% MeCN, 0.1% formic acid). The LTQ-Orbitrap Velos system acquired full scan survey spectra (*m/z* 350 to 1500) with a 15000 resolution at m/z 400. A maximum of the 10 most abundant multiple charged ions registered in each survey spectrum were selected in a data-

dependent manner, fragmented by collision induced dissociation (multistage activation enabled for neutral loss of phosphate group) with a normalized collision energy of 35% and scan in the LTQ ion trap (*m/z* 50 to 2000). In the data-dependent acquisition, a dynamic exclusion was enabled (exclusion list restricted to 500 entries, 60 s duration and 10 ppm mass window).

Phosphopeptide identification was performed by uploading the raw data to Mascot Distiller v2.3.2.1, which was used to smoothen and centroid the MS/MS data. Mascot v2.3.0 search engine was used to match peaks to peptides in proteins present in the WormPep233 FASTA file (downloaded January 2013) containing 26011 sequences. The process was automated with Mascot Daemon v2.3 with MS1 (precursor ions) mass tolerance set to 10 ppm and MS2 (product ions) mass tolerance set to 600 mmu. Carbamidomethylation of cysteine residues was assigned as a fixed modification whereas phosphorylation of serine, threonine and tyrosine residues, pyro-glutamination on N-terminal glutamine residues, and oxidation on methionine residues were assigned as variable modifications. Trypsin was selected as the digestion enzyme (C-terminal arginine and lysine cleavage expected provided there is not a proline C-terminal to the arginine/lysine residue) and 2 missed cleavages were allowed. Sites of modification are reported when they had Mascot delta scores >10. Delta scores were calculated as previously described (*Savitski et al., 2011*). Otherwise the site of modification was deemed to be ambiguous and phosphopeptides are reported as the start-end residues within the protein sequence.

## Acknowledgements

We are grateful to the *Caenorhabditis* Genetics Center for providing strains and to M Zetka for the HTP-3 antibody. We thank members of the Colaiácovo lab for critical reading of this manuscript and providing helpful suggestions. We thank S Arur and T Schedl for data showing expression of RHO-1 in the germ line and decreased dpMPK-1 upon partial depletion of *rho-1* in the germline as well as for critical reading of this manuscript. This work was supported by National Institutes of Health grant R01GM086434 to MB, R01GM085150 to TS, an MRC-core funded grant to EMP, AG011085 to JWH, a fellowship from the Human Frontier Science Program to YBT, a fellowship from the Lalor Foundation to SN and R01GM072551 to MPC.

## Additional information

### Funding

| Funder | Grant reference number | Author |
| --- | --- | --- |
| The Lalor Foundation | | Saravanapriah Nadarajan |
| Human Frontier Science Program | | Yonatan B Tzur |
| National Institutes of Health | R01GM086434 | Michael D Blower |
| MRC-Core funding | | Enrique Martinez-Perez |
| National Institutes of Health | AG011085 | J Wade Harper |
| National Institutes of Health | R01GM072551 | Monica P Colaiacovo |

The funders had no role in study design, data collection and interpretation, or the decision to submit the work for publication.

### Author contributions

SN, Conception and design, Acquisition of data, Analysis and interpretation of data, Drafting or revising the article; FM, Acquisition of data, Analysis and interpretation of data, Drafting or revising the article; YBT, EM-P, Drafting or revising the article, Contributed unpublished essential data or reagents; NF, OC, AM, PF, APS, PRC, AJ, MDB, Acquisition of data, Contributed unpublished essential data or reagents; JWH, Analysis and interpretation of data, Contributed unpublished essential data or reagents; MPC, Conception and design, Analysis and interpretation of data, Drafting or revising the article

## Author ORCIDs

Oliver Crawley, http://orcid.org/0000-0002-5054-0051
Pedro R Cutillas, http://orcid.org/0000-0002-3426-2274

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
