## [Decision Letter]

Thank you for submitting your work entitled "The MAP kinase signaling pathway coordinates crossover designation with synaptonemal complex disassembly during meiosis" for consideration by *eLife*. Your article has been reviewed by 3 peer reviewers, one of whom, Bernard de Massy, is a member of our Board of Reviewing Editors, and the evaluation has been overseen by a Reviewing Editor and James Manley as the Senior Editor.

The reviewers have discussed the reviews with one another and the Reviewing editor has drafted this decision to help you prepare a revised submission.

Summary:

The study by Nadarajan et al. entitled "The MAP kinase signaling pathway coordinates crossover designation with synaptonemal complex disassembly during meiosis" begins with the discovery that *ect-2*, a regulator of the MAP kinase signaling pathway, is required for AIR-2 recruitment to/stabilization on bivalent chromosomes at the end of meiotic prophase in the *C. elegans* germline. Consistent with the possibility that ECT-2 regulates MAP kinase signaling in the *C. elegans* germline, the study also found that an activated form of MPK-1 (dbMPK-1) is reduced in mid/late prophase regions of the *ect-2* -deficient germlines, and that a let-60 gain-of-function allele is epistatic to an *ect-2* reduction-of-function allele with respect to a meiotic chromosomal phenotype described later in the study (SC disassembly, see below).

Although *ect-2* became of interest based on the fact that it is involved in AIR-2 localization to meiotic chromosomes, this study does not go on to characterize this potential function of MAP kinase signaling in the *C. elegans* germline. Instead the authors are interested in the possibility of a functional relationship between MAP kinase signaling and the asymmetric disassembly of synaptonemal complex (SC), a meiotic chromosomal structure that is assembled at the interface of chromosome axes along the full length of the bivalent. This relationship is being analyzed by various approaches.

The major findings presented in this study are thorough and strong and make the clear case that MAP kinase signaling can influence SC disassembly dynamics during late meiotic prophase in *C. elegans* germlines. However, the interpretations of several key experiments highlighted in the title and Abstract are not convincing, and the manuscript needs substantial revision in order to provide a clear description and understanding of the pathway analyzed.

Essential revisions:

1) The data presented fall short of supporting the authors' assertion that MAP kinase signaling is part of "a mechanism set in place to trigger SC disassembly". If MAP kinase signaling is part of an essential mechanism that normally triggers SC disassembly, one could expect to see misregulated SC disassembly in *ect-2* and *mpk-1* loss-of-function mutants. Alternatively, the role of MAP kinase may be only inhibitory (at least when constitutively activated) and its role in a wild type context with respect to SC disassembly thus remains undetermined.

2) Interpretation of accumulation of SYP-1 as evidence for failure of SC disassembly (or abrogation of, as in the Abstract) is very confusing. Constitutive phosphorylation of SYP-2 is correlated with persistence of SYP-1 on long arms but not with failure to disassemble the SC otherwise homolog arms would be kept aligned. Thus SC does disassemble in these conditions but some properties of SYP-1 is altered leading to its retention on axis.

3) Direct evidence for in vivo phosphorylation of Syp-2 by Map kinase is not convincing: the *mpk-1* null mutants do not progress from early/mid to late pachytene (as pointed out in the first paragraph of the subsection “The ERK MAP kinase pathway is involved in regulating SC dynamics”). Thus, if (hypothetically) SYP-2 is only capable of being phosphorylated during late pachytene, an absence of SYP-2 phosphorylation in *mpk-1* null mutants could be a consequence of a progression defect, and not because SYP-2 is normally a direct target of MAP kinase activity in vivo.

4) Only SYP-1 is being monitored for SC disassembly, the other SC components should also be tested. Whether they are interdependent or not in the context analyzed here is not known.

5) The conclusion for a role of MPK for coordinating crossing over designation and SC disassembly requires the validation of the prediction that double mutants as *cosa-1 mpk-1* (rf), *cosa-1 ect-2*(rf), and *cosa-1 syp-2*[S25A] show wild type SC disassembly at late pachytene/early diplotene.

6) The authors propose that the MAP kinase coordination is essential for ensuring accurate chromosome segregation. However, the consequences of failure to disassemble the SC on the long arm are not clear? Is the second (i.e. short arm disassembly) impacted? Why do SC disassembly defective mutants have no increase of% of males? (Table 1).

7) How are AIR-2 and LAB-1 localized at diakinesis in the *syp-2* phosphomimetic mutant?

Specific comments:

1) Figures of diplotene (2, 3, 5 and Figure 2—figure supplement 2): A general comment is that the quality of several images are poor and the actual identification of short and long arms difficult to sort out. Fundamentally, one of the key phenotype evaluated (SC disassembly or not) is based on the colocalization (or not) of *Htp-3* and *Syp-1*.

However, no quantification is provided on these colocalizations, per nucleus, and how many nuclei have been looked at, at which stage, which criteria were used for staging and what is the variability of phenotypes? In terms of staging for instance in Figure.2A *lip-1*(rf), the nucleus looks like diakinesis with arms separated? Overall providing one nucleus is not sufficient. Quantification is needed. This should be revised.

Specific issues on Figure 2:

Persistence of SC on long arms in *ect-2gf* looks much more robust than persistence of SC on long arms in *let-60gf* where co-localization between *Htp-3* and *Syp-1* is not clear.

*Mpk-1(lf)* mutant does not look like diplotene, probably earlier prophase, but the authors write in the text that SC disassembly is normal in this mutant? This is not convincing. In addition, the conditions used for analyzing the temperature sensitive mutants should be clarified and given for each experiment with wild type controls at both permissive and non-permissive temperatures: Previous studies have documented SC polycomplex formation in wild-type germlines at 25 degrees, among other defects. Even though it is mentioned for Table 1, it is not clearly mentioned in the Materials and methods that the WT worms were subject to the same temperature shifts. No data is presented that would give an assessment of how quickly the alleles are inactivated or potentially even re-activated. This would be crucial to know, especially when analyzing the *ect-2 rf* mutant, that has only subtle phenotypes. In addition, a *let-60gf* allele is used that is not the standard allele *let-60(n1046)*, without any explanation or reassurance that it behaves as the standard allele.

Similar comment to *ect-2 mpk-1* double mutant with distinct DAΠ and *Htp-3* staining patterns.

Does *ect-2 gf* have a phenotype with respect to axis morphology (HTP-3) or chromosome structure (Figure 1)?

2) Figure 4:

It should be mentioned in the text, legend and figure that it is not the endogenous SYP that is detected here. Please clarify which experiments involve SYP-2::GFP.

Other comments:

1) Figure 4: Please clarify: the legend does not clearly indicate what is shown in the figure. The numbers in orange, blue and red, "b ions" and "y ions" and the plot itself are not easy to understand.

2) Schematic representation (such as on Figure 2, Figure 3 and 11) can help but should be also provided for all genotypes.

3) Is the S25 phosphorylation site of *Syp-2* conserved?

4) Size markers and loading controls are needed on all western blots. For instance, it is impossible to know whether the bands detected in 4D have the same migration as the ones in 4E (upper? or lower?). Provide a longer exposure for Figure 4.

In addition, the *Syp-2* band in *mpk-1* (null) has a slower migration? Is this reproducible? Does it mean an *Mpk-1* independent phosphorylation? In addition, with respect to migration, the same migration does not allow concluding that the protein is not phosphorylated (contrary to what is written in the text).

5) In the first paragraph of the subsection “Phosphorylation of SYP-2 is dependent on the ERK MAP kinase pathway”: You may mean mutant lysates?

6) Table 1: Why is *let-60(ga)* epistatic to *ect-2(ax)* for brood size and not for embryonic lethality?

In *ect-2(zh8); mpk-1(ga111)*, 0% hatch and 0% embryonic lethality. Should it be N/A instead? Why does *ect-2 (ax)* have a HIM phenotype? Why not other mutants? Is the suppression of the Him phenotype in this background (*ect-2(ax) let-60(ga)*) stat. significant?

For brood size indicate if the interval is based on SD. SD cannot be used on percentages.

Is% of HIM the percentage of male progeny or (less likely) the percentage of P0 worms producing males. Please clarify. The 3.25% of males in *ect-2 (ax)* seems to correspond to an average of 1 male per brood. Is this significantly different from 0? Statistical analysis should be provided to validate the conclusions.

7) ECT-GFP, Figure 1—figure supplement 3:

Statistical analysis for suppression is missing. Is it not significant? The resolution of ECT-2::GFP images is not of high enough quality to be able to conclude that ECT-2 may be nuclear. Magnifications of diplotene and diakinesis nuclei would be valuable with negative control not expressing ECT::GFP.

8) Why is data shown for *ect-2 rf* and not of *ect-2* (RNAi), given that only 50% of *ect-2 rf*, but 100% of RNAi showed AIR-2 "recruitment" defect?

9) Figure 2—figure supplement 7:

The expression pattern for ECT-2 and RHO-1 is similar – is there co-localization?

10) How are *Syp-1, Syp-2* and *Syp-3* levels impacted in mutants defective for SC disassembly? Could IF observation result at least in part from the accumulation of SYP proteins (on long and may be also on short arms?)

11) Figure 4–Figure 6:

The analysis of *ect* mutants with regard to SYP-2 phenotypes is missing. Why is this not discussed?

12) for Figure 2—figure supplement 5 and Figure 2—figure supplement 6, it would be nice to have a ratio of the DAPI:dpMPK1 signal to support the conclusion that dpMPK1 levels are altered.

---

## [Author Response]

*The major findings presented in this study are thorough and strong and make the clear case that MAP kinase signaling can influence SC disassembly dynamics during late meiotic prophase in C. elegans germlines. However, the interpretations of several key experiments highlighted in the title and Abstract are not convincing, and the manuscript needs substantial revision in order to provide a clear description and understanding of the pathway analyzed.*

*1) The data presented fall short of supporting the authors' assertion that MAP kinase signaling is part of "a mechanism set in place to trigger SC disassembly". If MAP kinase signaling is part of an essential mechanism that normally triggers SC disassembly, one could expect to see misregulated SC disassembly in ect-2 and mpk-1 loss-of-function mutants. Alternatively, the role of MAP kinase may be only inhibitory (at least when constitutively activated) and its role in a wild type context with respect to SC disassembly thus remains undetermined.*

As mentioned in the text, ECT-2 and MAP kinase also have roles in the pre-meiotic region of the germline and meiotic progression, respectively. This impedes the analysis of SC disassembly in *ect-2* and *mpk-1* null mutants, and is the reason why we utilized the available temperature sensitive mutants instead:

“Importantly, we shifted *ect-2(ax751rf)* worms to the non-permissive temperature at the L4 larval stage to bypass the requirements for ECT-2 during somatic development and germ cell mitotic proliferation such as seen in *ect-2(e1778)* null mutants, which are sterile and exhibit fewer germ cells with abnormal nuclei (Figure 1—figure supplement 2).”

“Since nuclei fail to progress from early/mid-pachytene to late-pachytene in *mpk-1(ga117)* null mutants [51], we analyzed SC disassembly in *mpk-1(ga111lf)* mutants.”

We found that the *ect-2(ax751)* temperature sensitive reduction-of-function mutant still has a reduced level of dpMPK-1 and it is still able to phosphorylate SYP-2 (Figure 2—figure supplement 5 and Figure 4). Therefore, we concluded that we were not able to see an SC disassembly defect in *ect-2(ax751)* mutants due to the presence of low level dpMPK-1 activity (Results, last paragraph).

The *mpk-1(ga111)* temperature sensitive loss-of-function mutant has been shown to still have a low level of dpMPK-1 activity and an incompletely penetrant pachytene arrest phenotype (Lee et al., 2007). SC disassembly still occurs in the nuclei that exit pachytene. The residual dpMPK-1 activity most likely also explains why we do not see an SC disassembly defect in this mutant (subsection “The ERK MAP kinase pathway is involved in regulating SC dynamics”, first paragraph).

*2) Interpretation of accumulation of SYP-1 as evidence for failure of SC disassembly (or abrogation of, as in the Abstract) is very confusing. Constitutive phosphorylation of SYP-2 is correlated with persistence of SYP-1 on long arms but not with failure to disassemble the SC otherwise homolog arms would be kept aligned. Thus SC does disassemble in these conditions but some properties of SYP-1 is altered leading to its retention on axis.*

SC disassembly has now been shown to be a two-step process in many organisms including budding yeast, female flies, worms and mice. In mice, for example, SYCP1, a central region component of the SC, is first lost from along the lengths of the chromosomes and retained at paired centromeres as cells progress towards metaphase. In a second step, SYCP1 is removed from centromeres by late diplotene and diakinesis (Bisig et al., 2012). In the case of *C. elegans*, first the SC disassembles from along the long arms of the bivalents and is restricted to the short arms where it is maintained through mid diakinesis. In a second step, it is then lost from the short arms by late diakinesis (around the -3 oocyte), as described in the text: “The SYP proteins are removed from the chromosomes in two steps, first SYPs are lost from the long arms of the bivalents in the late pachytene to early diplotene stage nuclei, but are selectively maintained on the short arms of the bivalents until late diakinesis (-3 oocyte; Figure 1).”

We have now also added a new diagram/cartoon to better illustrate this process at the beginning of the manuscript (Figure 1).

Based on the two-step process of SC disassembly it is correct to describe what we are seeing as defects in the ability of the SC to disassemble from along the long arms of the bivalents. We understand the reviewer’s point that since the SC is built to hold homologs together, then normal disassembly equates to the physical separation of homolog arms. However, the term SC disassembly has been widely utilized to refer to the physical separation of homolog arms and the loss of the SC proteins from specific chromosome subdomains. The latter, however, is not occurring normally when there is constitutive phosphorylation of SYP-2. To clarify this terminology, we now explain it the first time that we describe the defect in our manuscript: “Those nuclei that had already gone through the leptotene/zygotene stage before the temperature shift showed normal synapsis during early and mid pachytene stages, but the central region components of the SC remained localized along the long arms of the bivalents and failed to become restricted to the short arms of the bivalents during late prophase (a defect herein referred to as impaired SC disassembly) (Figure 2)”.) We also replaced the term “abrogates” with “impairs” in the two locations in the text where that term was utilized. It is also important to note that in *C. elegans* the central region of the SC is comprised of four SYP proteins (SYP-1/2/3/4). These four SYP proteins have been shown before to be interdependent for their localization (Colaiacovo et al., 2003; Smolikov et al., 2007 and Smolikov et al., 2009). Therefore, the localization of SYP-1 reflects the localization pattern of all the SYPs and previous publications from many labs, including ours, have used SYP-1 as a readout to study the SC assembly and disassembly process (Nabeshima et al., 2005; Bhalla et al., 2008; Martinez-perez et al., 2008; Clemon et al., 2013). However, to further clarify this we now also include analysis of SYP-2, SYP-3 and SYP-4 (Figure 5—figure supplement 2) and show that all four SYP proteins fail to be lost from along the long arms. This further indicates that the presence of SYP-1 on the long arm at a stage when it should have normally been lost from this chromosome subdomain is not due to any altered properties of SYP-1 and instead indicates that there is a defect in disassembling all SC central region components from along the long arms of the bivalents.

*3) Direct evidence for in vivo phosphorylation of Syp-2 by Map kinase is not convincing: the mpk-1 null mutants do not progress from early/mid to late pachytene (as pointed out in the first paragraph of the subsection “The ERK MAP kinase pathway is involved in regulating SC dynamics”). Thus, if (hypothetically) SYP-2 is only capable of being phosphorylated during late pachytene, an absence of SYP-2 phosphorylation in mpk-1 null mutants could be a consequence of a progression defect, and not because SYP-2 is normally a direct target of MAP kinase activity in vivo.*

The activated form of MPK-1(dpMPK-1) is present starting from mid-pachytene in wild type (Lee et al., 2007), and this led us to hypothesize that SYP-2 could get phosphorylated starting from mid-pachytene (not late pachytene as indicated by the reviewer). In the *mpk-1(ga117)* null mutant, even though nuclei do not progress to late pachytene, they still have early and mid-pachytene stage nuclei as mentioned in the manuscript: “Since nuclei fail to progress from early/mid-pachytene to late-pachytene in *mpk-1(ga117)* null mutants [Lee et al., 2007]”

In the Results section we stated: “This could be due to only a subset of the SYP-2 protein being phosphorylated by MPK-1. This is consistent with the expression pattern of MPK-1 in the germline where the active form of dpMPK-1 is expressed only from mid to late pachytene. Meanwhile, SYP-2 is expressed from the leptotene/zygotene stage to late diakinesis. Therefore, SYP-2 could be phosphorylated by dpMPK-1 at the mid to late-pachytene region thereby regulating the localization of SYP-2 in a spatio-temporal manner.”

We have also provided several lines of evidence that show that SYP-2 is a direct target of MAP kinase: 1) We identified the SYP-2 (S25) MAP kinase phosphorylation site by two different approaches in the worm (pull downs followed by mass spectrometry and by a phosphoproteomics approach); 2) We used CIP assays and the *mpk-1* null mutant to show that it is a target; 3) in vitro kinase assays showed that the S25 site is phosphorylated by MAP kinase; 4) Analysis of a phosphomimetic mutant in which the syp-2(S25) site was mutated to *syp-2(S25D),* showed in vivo that the mutant phenocopies the *ect-2(gf)* and MAP kinase gain-of-function phenotypes consistent with our hypothesis. We attempted to generate a phospho-specific antibody against the SYP-2(S25) site but we were not successful.

*4) Only SYP-1 is being monitored for SC disassembly, the other SC components should also be tested. Whether they are interdependent or not in the context analyzed here is not known.*

To address the reviewers’ comment we also analyzed SYP-2, SYP-3 and SYP-4 localization in *ect-2(gf), let-60(gf),* and *syp-2* phosphomimetic mutants. Consistent with our previous findings for SYP-1, all other three SYP proteins also failed to disassemble normally from along the long arms of the bivalents in *ect-2(gf), let-60(gf)* and the *syp-2* phosphomimetic mutant indicating an SC disassembly defect. This new data is now shown as Figure 5—figure supplement 2).

*5) The conclusion for a role of MPK for coordinating crossing over designation and SC disassembly requires the validation of the prediction that double mutants as cosa-1 mpk-1 (rf), cosa-1 ect-2(rf), and cosa-1 syp-2[S25A] show wild type SC disassembly at late pachytene/early diplotene.*

As per the reviewers’ suggestion, we analyzed SC disassembly in *cosa-1(me13) mpk-1(ga111), ect-2(ax751); cosa-1(me13)* and *cosa-1(me13);syp-2(S25A)* double mutants. Consistent with our conclusion that MAP kinase coordinates crossing over designation with SC disassembly, we found that the SC is able to disassemble in these double mutants, but with different penetrance levels. Specifically, *mpk-1(ga111)* is able to completely suppress the SC disassembly defect observed in a *cosa-1(me13)* mutant, whereas *ect-2(ax751)* and *syp-2(S25A)* show a 11.5% and 40.9% suppression of the *cosa-1(me13)* SC disassembly defects, respectively. Since *ect-2(ax751)* is a reduction-of-function mutant and it has residual dpMPK-1 activity (Figure 2—figure supplement 5 and Figure 4), we would not expect a complete suppression. We hypothesize that the reason we see a complete suppression in *cosa-1(me13) mpk-1(ga111),* but a partial suppression in *cosa-1(me13);syp-2(S24A)* double mutants, is that another SYP protein could be a substrate of MPK-1. However, we did not find any additional MAP Kinase phosphorylation sites in any of the SYP proteins, other than the S25 site in SYP-2. Interestingly though, there are phosphorylation sites for polo-like kinase on SYP-1 and SYP-4, which were verified by mass spectrometry. Polo-like kinase itself has been identified as a target of MPK-1 in *C. elegans* (Arur et al., 2009). It is possible that MAP kinase prevents the SC from disassembling prematurely through direct phosphorylation of SYP-2 and phosphorylation of SYP-1 and SYP-4 by polo-like kinase. The analysis of all the requested double mutants and interpretation of the observed outcomes have been included in Figure 6 and in the main text (Results section, last paragraph).

*6) The authors propose that the MAP kinase coordination is essential for ensuring accurate chromosome segregation. However, the consequences of failure to disassemble the SC on the long arm are not clear? Is the second (i.e. short arm disassembly) impacted? Why do SC disassembly defective mutants have no increase of% of males? (Table 1).*

We cannot detect an increase in the% of male progeny because *ect-2(gf)* and *let-60(gf)* mutants have really reduced brood sizes (a very low number of eggs are laid), mostly in the range of 0.7 to 4.5 eggs are laid per hermaphrodite in these mutants, instead of the close to 200 eggs laid per hermaphrodite by wild type, as shown in Table 1, and this masks the ability to detect males. However, in the *syp-2* phosphomimetic mutant, where we see an SC disassembly defect only in 54% of the animals (as shown in Figure 5), we see a milder reduction in brood size (109 ± 32), accompanied by 14.3% embryonic lethality and 2.1% males, indicating that impaired SC disassembly is resulting in a defect in accurate chromosome segregation. We now include this information in the subsection “Phosphorylation of SYP-2 at S25 prevents SC disassembly”. We also suggest that SC disassembly from the short arm could be regulated either by different kinases or by a different mechanism (as described above in response to point #5 and in the manuscript). In the second step, given the interdependence of the SYP proteins, the SC would end up being removed by these other kinases (potentially polo-like kinases) from both long and short arms. We are currently examining the regulation of the second step of SC disassembly and its communication with the first step of SC disassembly, but these studies fall beyond the scope of this current manuscript and will be described in detail elsewhere.

7) How are AIR-2 and LAB-1 localized at diakinesis in the syp-2 phosphomimetic mutant?

We have examined this and show that both LAB-1 and AIR-2 localization in the *syp-2* phosphomimetic mutant is similar to wild type. This new data is now shown in Figure 5—figure supplement 2. This supports the specificity of MPK-1 operating on SYP-2 in the regulation of disassembly from the long arms of the bivalents.

*Specific comments:*

*1) Figures of diplotene (2, 3, 5 and Figure 2—figure supplement 2): A general comment is that the quality of several images are poor and the actual identification of short and long arms difficult to sort out. Fundamentally, one of the key phenotype evaluated (SC disassembly or not) is based on the colocalization (or not) of Htp-3 and Syp-1.*

To address this we have provided better resolution images for these figures and included higher magnifications images of a single bivalent to better illustrate what is being observed along the long and short arms of the bivalents. We also included diagrams/cartoons to make this clear. Finally, we also included additional images where the long arms are marked with LAB-1 for key mutants as a supplemental figure (Figure 5—figure supplement 3).

*However, no quantification is provided on these colocalizations, per nucleus, and how many nuclei have been looked at, at which stage, which criteria were used for staging and what is the variability of phenotypes? In terms of staging for instance in Figure 2 lip-1(rf), the nucleus looks like diakinesis with arms separated? Overall providing one nucleus is not sufficient. Quantification is needed. This should be revised.*

As recommended, we now included histograms showing the percentage of nuclei and the number of nuclei scored for each genotype in Figure 2 and Figure 5 to show the variability of the phenotypes and we indicate the stage and criteria used for staging in each figure legend.

*Specific issues on Figure 2: Persistence of SC on long arms in ect-2gf looks much more robust than persistence of SC on long arms in let-60gf where co-localization between Htp-3 and Syp-1 is not clear.*

We treated the *ect-2(gf)* with same intensity parameters as all other images. We replaced the previous *let-60(gf)* image with a better representative that shows the phenotype more clearly. We have now also included insets with a higher magnification view of a single bivalent accompanied by a diagram for better ease of visualization of the phenotypes.

*Mpk-1(lf) mutant does not look like diplotene, probably earlier prophase, but the authors write in the text that SC disassembly is normal in this mutant? This is not convincing.*

We have now replaced this image with a clear diplotene stage nucleus observed in this mutant. As mentioned above, we now also include further magnified insets, a diagram, and quantification, which show that the SC localization is restricted to the short arm in the *mpk-1(lf)* temperature-sensitive loss-of-function mutant.

*In addition, the conditions used for analyzing the temperature sensitive mutants should be clarified and given for each experiment with wild type controls at both permissive and non-permissive temperatures: Previous studies have documented SC polycomplex formation in wild-type germlines at 25 degrees, among other defects. Even though it is mentioned for Table 1, it is not clearly mentioned in the Materials and methods that the WT worms were subject to the same temperature shifts. No data is presented that would give an assessment of how quickly the alleles are inactivated or potentially even re-activated. This would be crucial to know, especially when analyzing the ect-2 rf mutant, that has only subtle phenotypes.*

We now explain in the Materials and methods that “The wild type worms used for comparisons with these mutants were all subjected to the same temperature shifts and examined at the same times as the mutants”. We also provide the conditions that were used for each experiment in each figure legend. Wild type animals were treated the same way as the mutants in every experiment. All of the ts alleles were analyzed 18-24 hrs post L4 at 25 degrees Celsius where the MAP kinase mutants have been clearly shown to exhibit phenotypes (Lee et al., 2007).

*In addition, a let-60gf allele is used that is not the standard allele let-60(n1046), without any explanation or reassurance that it behaves as the standard allele.*

Lee et al., 2007 have shown that the *let-60(n1046)* mutant exhibits a wild type dpMPK-1 pattern whereas *let-60(ga89)* is a gain-of-function mutant that shows constitutive activation of dpMPK-1 in the germline and is therefore the appropriate allele to use in our studies.

*Does ect-2 gf have a phenotype with respect to axis morphology (HTP-3) or chromosome structure (Figure 1)?*

We now included higher magnification insets for Figure 1 and Figure 2, which show that the *ect-2(gf)* mutant does not exhibit any obvious defect in axis morphology.

*2) Figure 4: It should be mentioned in the text, legend and figure that it is not the endogenous SYP that is detected here. Please clarify which experiments involve SYP-2::GFP.*

In addition to the original mention in the Material and Methods section we now also indicate this more clearly in the text, the figure legend, and the figure itself (subsection “Immunoprecipitation” and subsection “Phosphorylation of SYP-2 is dependent on the ERK MAP kinase pathway“, second paragraph).

*Other comments:*

*1) Figure 4: Please clarify: the legend does not clearly indicate what is shown in the figure. The numbers in orange, blue and red, "b ions" and "y ions" and the plot itself are not easy to understand.*

We have changed the figure legend to clarify this. It now reads as follows: “(B) MS/MS fragmentation spectrum for SYP-2 phosphopeptide VSFASPVSSSQK in the range 100-1300m/z. The annotated spectrum shows fragment ion species matched between theoretical and measured values. “b ions” are generated through fragmentation of the peptide bond from the N-terminus, whereas “y ions” are generated through fragmentation from the C-terminus. Ion species detected with a mass loss of 98 (phosphoric acid) are indicated in yellow; those ions without phospho-loss are annotated in red. Analysis of “y” and “b” ions with and without phospho-loss is consistent with phosphorylation of the second serine (VSFApSPVSSSQK), corresponding to S25 of the SYP-2 protein.”

*2) Schematic representation (such as on Figure 2, Figure 3 and 11) can help but should be also provided for all genotypes.*

As suggested, we now show schematic representations for all the genotypes

analyzed.

*3) Is the S25 phosphorylation site of Syp-2 conserved?*

As shown in Figure 7—figure supplement 1 and described in the subsection “Regulation of asymmetric SC disassembly – an evolutionarily conserved mechanism?”, the presence of a MAP kinase site is conserved for some of the SC proteins in both mice and humans, namely SYCE1, SYCE2 and TEX12, but the position of the phosphorylation site is different, with TEX12 being the most similar by carrying an S27 phosphorylation site.

*4) Size markers and loading controls are needed on all western blots. For instance, it is impossible to know whether the bands detected in 4D have the same migration as the ones in 4E (upper? or lower?). Provide a longer exposure for Figure 4.*

We now show loading controls for all western blots. Size markers are now also shown for Figure 4. However, Figure 4 are phos-tag gels used for better resolution of the bands. Size markers are not recommended for phos-tag gels since they can cause problems in uniform band migration, so here we now show the expected sizes. The Western blot on Figure 4 was run on a polyacrylamide gradient gel to show in vivo interaction between SYP-2 and dpMPK-1. Unlike the phos-tag gel, we were not able to see the separation of phosphorylated versus non-phosphorylated SYP-2 proteins in the polyacrylamide gel so we cannot say whether the band that pulls down with dpMPK-1 is a phosphorylated or non-phosphorylated band. The data originally shown in Figure 4 has now been replaced by a longer exposure for a new blot given that we were also asked to also analyze the *ect-2* null, rf and gf mutants (see Reviewer comment #11) and that is now shown in Figure 4.

*In addition, the Syp-2 band in mpk-1 (null) has a slower migration? Is this reproducible? Does it mean an Mpk-1 independent phosphorylation? In addition, with respect to migration, the same migration does not allow concluding that the protein is not phosphorylated (contrary to what is written in the text).*

This was just an artifact of how samples ran in this particular gel. We have confirmed that there is no slower migrating SYP-2 band in the *mpk-1* null mutant through 8 independent repeats of this experiment and analysis. We now provide a blot that is a more accurate representation of what we observe. With respect to migration, if the SYP-2 protein in the phosphodead mutant runs at the same size as the SYP-2 band observed in a *mpk-1* null mutant that indicates that SYP-2 is most likely not phosphorylated in the *mpk-1* null mutant.

*5) In the first paragraph of the subsection “Phosphorylation of SYP-2 is dependent on the ERK MAP kinase pathway”: You may mean mutant lysates?*

These were wild type lysates.

*6) Table 1: Why is let-60(ga) epistatic to ect-2(ax) for brood size and not for embryonic lethality?*

*In ect-2(zh8); mpk-1(ga111), 0% hatch and 0% embryonic lethality. Should it be N/A instead? Why does ect-2 (ax) have a HIM phenotype? Why not other mutants? Is the suppression of the Him phenotype in this background (ect-2(ax) let-60(ga)) stat. significant?*

*For brood size indicate if the interval is based on SD. SD cannot be used on percentages.*

*Is% of HIM the percentage of male progeny or (less likely) the percentage of P0 worms producing males. Please clarify. The 3.25% of males in ect-2 (ax) seems to correspond to an average of 1 male per brood. Is this significantly different from 0? Statistical analysis should be provided to validate the conclusions.*

We have modified this table and expanded its legend to clarify the issues raised by the reviewer. Regarding the epistasis question for *let-60(ga89)* and *ect-2(ax751)*, this is an issue of reduced brood size. *ect-2(ax751)* mutants have good brood sizes with each P0 laying on average 192 ± 32.3 eggs (the interval is based on standard deviation because the brood size shown is not a percentage), whereas brood sizes for *let-60(ga89)* mutants are on average3.1 ± 5.6. As recommended we now use N.A. where no eggs were laid and therefore the Emb and Him phenotypes cannot be scored. We can score a Him phenotype for *ect-2(ax751)* but not for the other mutants because this *ect-2* mutant has a good brood size while all the other mutants have really reduced brood sizes that mask the Him phenotype (the very low numbers of eggs and viable progeny reduce the probability of seeing any males). We now explain how the% of males is calculated in the legend for this table.

*7) ECT-GFP, Figure 1—figure supplement 3: Statistical analysis for suppression is missing. Is it not significant? The resolution of ECT-2::GFP images is not of high enough quality to be able to conclude that ECT-2 may be nuclear. Magnifications of diplotene and diakinesis nuclei would be valuable with negative control not expressing ECT::GFP.*

Yes, suppression of the *ect-2(rf)* phenotype by ECT-2::GFP is statically significant. We have now incorporated this statistical analysis into Figure 1—figure supplement 3 and its figure legend. As suggested, we now also included high magnification images showing nuclear localization in diplotene and diakinesis stage nuclei in Figure 1—figure supplement 4 and we also show the negative control not expressing ECT-2::GFP.

*8) Why is data shown for ect-2 rf and not of ect-2 (RNAi), given that only 50% of ect-2 rf, but 100% of RNAi showed AIR-2 "recruitment" defect?*

We could not analyze the *ect-2(RNAi)* further, and instead used the *ect-2(rf)* mutant, because *ect-2(RNAi)* gives a disorganized germline similar to the null mutant.

*9) Figure 2—figure supplement 7: The expression pattern for ECT-2 and RHO-1 is similar – is there co-localization?*

We examined this prompted by the reviewer’s question. Please see new Figure 1—figure supplement 4 for co-localization between ECT-2 and RHO-1 in the wildtype germline. We see complete co-localization between ECT-2 and RHO-1 in the germline.

*10) How are Syp-1, Syp-2 and Syp-3 levels impacted in mutants defective for SC disassembly? Could IF observation result at least in part from the accumulation of SYP proteins (on long and may be also on short arms?)*

Western blot analysis of SYP-2 and SYP-3 proteins in different mutant backgrounds including *ect-2(gf)* and *let-60(gf)* does not show significant difference in protein level between mutants and wild type suggesting that the SC disassembly defect in the mutants is not due to accumulation of SYP proteins (Figure 4 and Figure 4—figure supplement 1).

*11) Figure 4–Figure 6: The analysis of ect mutants with regard to SYP-2 phenotypes is missing. Why is this not discussed?*

We have now included western blot analysis showing SYP-2 in *ect-2(null), ect-2(rf)* and *ect-2(gf)* mutants in Figure 4. Consistent with our previous data, there is no SYP-2 phosphorylation in the *ect-2* null mutant and we see phosphorylated as well as non-phosphorylated SYP-2 bands in both *ect-2(rf)* and *ect-2(gf)* mutants (Figure 4). This information is now also discussed in the main text.

12) for Figure 2—figure supplement 5 and Figure 2—figure supplement 6, it would be nice to have a ratio of the DAPI:dpMPK1 signal to support the conclusion that dpMPK1 levels are altered.

We quantified the dpMPK-1 fluorescence intensity in *ect-2(rf)* and *ect-2(gf)* mutants compared to wild type and we now show this in Figure 2—figure supplement 5. This quantification supports the conclusion that dpMPK-1 levels are altered in these mutants.